



# Estimating the uncertainty of the greenhouse gas emission accounts in Global Multi-Regional Input-Output analysis

Simon Schulte[1, *], Arthur Jakobs[2], and Stefan Pauliuk[1]

[1]Industrial Ecology Group, University of Freiburg, 79183 Freiburg, Germany
[2]Laboratory for Energy Systems Analysis, Paul Scherrer Institut, 5232 Villigen, Switzerland

**Correspondence:** Simon Schulte (simon.schulte@indecol.uni-freiburg.de)

**Abstract.**

Global multi-regional input-output (GMRIO) analysis is the standard tool to calculate consumption-based carbon accounts at the macro level. Recent inter-database comparisons have exposed discrepancies in GMRIO-based results, pinpointing greenhouse gas (GHG) emission accounts as the primary source of variation. A few studies have delved into the robustness of GHG emission accounts, using Monte-Carlo simulations to understand how uncertainty from raw data propagates to the final GHG emission accounts. However, these studies often make simplistic assumptions about raw data uncertainty and ignore correlations between disaggregated variables.

Here, we compile GHG emission accounts for the year 2015 according to the resolution of EXIOBASE v3, covering $CO_2$, $CH_4$ and $N_2O$ emissions. We propagate uncertainty from the raw data, namely the United Nations Framework Convention on Climate Change (UNFCCC) and EDGAR inventories, to the GHG emission accounts, and then further to the GHG footprints. We address both limitations from previous studies. First, instead of making simplistic assumptions, we utilise authoritative raw data uncertainty estimates from the National Inventory Reports (NIR) submitted to the UNFCCC and a recent study on uncertainty of the EDGAR emission inventory. Second, we account for inherent correlations due to data disaggregation by sampling from a Dirichlet distribution.

Our results show a median coefficient of variation (CV) for GHG emission accounts at the country level of 4% for $CO_2$, 12% for $CH_4$, and 33% for $N_2O$. For $CO_2$, smaller economies with significant international aviation or shipping sectors show CVs as high as 94%, as seen in Malta. At the sector level, uncertainties are higher, with median CVs of 94% for $CO_2$, 100% for $CH_4$, and 113% for $N_2O$. Overall, uncertainty decreases when propagated from GHG emission accounts to GHG footprints, likely due to the cancelling out effects caused by the distribution of emissions and their uncertainties across global supply chains. Our GHG emission accounts generally align with official EXIOBASE emission accounts and OECD data at the country level, though discrepancies at the sectoral level give cause for further examination.

We provide our GHG emission accounts with associated uncertainties and correlations at https://doi.org/10.5281/zenodo.10041196 (Schulte et al., 2023) to complement the official EXIOBASE emission accounts for users interested in estimating the uncertainties of their results.



# 1 Introduction

## 1.1 Problem setting

Currently, most climate policy focuses on greenhouse gas (GHG) emissions that physically occur within the geographical boundaries of a country (Steininger et al., 2016). This territorial perspective, however, neglects both, emissions caused in international territories such as from international air transport, as well as emissions embodied in trade, which due to globalisation nowadays constitute a major share in the life cycle impacts of most country's consumption (Peters et al., 2011; Pan et al., 2017; Hertwich and Wood, 2018). From an equity perspective, this approach can obscure the environmental responsibility of countries that outsource production and thus emissions to other nations, potentially placing a disproportionate burden on countries where production takes place while benefiting from the consumed goods. To complement the territorial perspective, the consumption-based perspective has been increasingly gaining attention in academia and the wider public in recent years (Tukker et al., 2020). At a macro level, those so-called consumption-based carbon accounts (CBCA) are typically calculated using environmentally-extended global multi-regional input-output (GMRIO) databases (Tukker et al., 2020). Those GMRIO databases usually consist of three building blocks: The inter-industry matrix, the final demand matrix, and the environmental satellite accounts (Miller and Blair, 2009). With GMRIO analysis, one can allocate those environmental impacts along global supply chains to the end consumers of products and services.

Although over the course of the last decade or so, GMRIO-based CBCAs have become a standard metric among academics, their adoption by policymakers remains limited compared to the territorial perspective (Tukker et al., 2018). One important reason for this restrained uptake is the lack of robust knowledge concerning model uncertainty. This absence of robust model uncertainty estimates poses a major challenge for decision-makers (Reale et al., 2017). Due to the complex nature of reality, our understanding of the effects of a decision will always be limited. Thus, making robust decisions in the real world, inevitably involves incorporating judgements regarding uncertainty (Lempert, 2003). For example, when choosing between two policy options, A and B, policymakers need not only understand that, *on average*, the modelled consequences of A surpass those of B, but also how *robust* those modelled results are. If, for instance, there's a 5% chance that A could lead to severe negative outcomes, a decision-maker might prefer B, even if the *average* expected result is more favourable for A.

Given the added complexity of GMRIO-based CBCA compared to territorial-based emission inventories, it becomes particularly crucial for the former to possess a profound understanding of uncertainties. Uncertainty in GMRIO modelling might arise from various sources. Here, we focus on parametric uncertainty referring to uncertainty in the outcome caused by uncertainty of input parameters (Huijbregts, 1998).

## 1.2 Literature review

A large part of environmentally-extended GMRIO studies make only, if at all, qualitative considerations of uncertainty (Zhang et al., 2019). Studies that include quantitative considerations of parametric uncertainties include Lenzen et al. (2010); Wilting (2012); Karstensen et al. (2015); Moran et al. (2018); Shrestha and Sun (2019); Zhang et al. (2019); Kanemoto et al. (2020); Abbood et al. (2023). Moreover, with Eora (Lenzen et al., 2013) and GLORIA (Lenzen et al., 2022) there exist two GMRIO





databases that publish uncertainty estimates in the form of standard deviations alongside each data entry, thus allowing the GMRIO practitioners to conduct an uncertainty analysis of their results by themselves.

All mentioned studies quantify parametric uncertainty by propagating uncertainty from model input parameters to model outputs via Monte-Carlo (MC) simulations. The first step in uncertainty propagation involves assigning probability distributions (mostly normal or log-normal distributions) to the model inputs. What is meant by model input differs depending on whether in the study an existing GMRIO database was used or whether a custom database was created. In case of the former, probability distributions are directly assigned to the input-output coefficients, i.e. the individual elements of the inter-industry,

final demand, or satellite extension matrices (Abbood et al., 2023; Shrestha and Sun, 2019; Kanemoto et al., 2020). In case of the latter, probability distributions are assigned to the raw data used to determine those coefficients (Lenzen et al., 2010; Karstensen et al., 2015; Zhang et al., 2019; Lenzen et al., 2013, 2022). Using MC simulations, i.e. repeatedly and randomly drawing raw data samples from those probability distributions, the uncertainty can be propagated either from the GMRIO database to the CBCA (in case of the former group of studies), or from the raw data needed to compile GMRIOs to the

GMRIO coefficients, and further to the CBCA (in case of the latter group of studies).

## 1.3    Research gaps

As we will argue in the following, all of the studies cited above have two major limitations: (1) First, all studies share a very simple modelling of the uncertainty of the raw input data. (2) Second, a disregard of correlations between variables obtained by disaggregating a common input data point.

(1) The raw data needed to compile GMRIO databases usually lacks quantitative information on uncertainty (Lenzen et al., 2013; Wilting, 2012). This holds true for most national IO tables, international trade statistics (such as BACI/Comtrade), macroeconomic data (e.g. GDP), or energy statistics. In the absence of quantitative uncertainty information, studies addressing uncertainty in GMRIO either estimate the raw data uncertainty using simple heuristics or model it through a power law regression.

In the first approach, simple heuristics are employed to assess the uncertainty of the raw data. This method is adopted by Wilting (2012); Abbood et al. (2023); Kanemoto et al. (2020); Shrestha and Sun (2019). For instance, Wilting (2012) applies heuristics such as: domestic input data being less uncertain than trade data, and sectors from one group being less uncertain than those from another, based on "differences in input characteristics." Through these heuristics, they determine an uncertainty estimate for each IO coefficient, that depends on the combination of the different characteristics.

The second approach involves using a statistical model, specifically a power law regression, to determine raw data uncertainties (Lenzen et al., 2010; Karstensen et al., 2015; Zhang et al., 2019). The authors fit a power law regression to some proxy data points using the size of the sector (in terms of emissions or financial volume) as the only predictor variable. The resulting power law relationship between the absolute sector size on the one hand, and uncertainty on the other, is then used to determine the uncertainties of the raw data (Lenzen et al., 2010). However, while the size of a sectoral flow (either financial or physical,

e.g. in the form of emissions) might explain some variability in the uncertainties between flows, there are likely other credible predictors one could, or perhaps should consider. This can also be observed in the poor fits of the power law regressions in





Lenzen et al. (2010): They report an $R^2$ of only 0.26 for financial transactions and 0.21 for carbon emissions, respectively, for their power law regressions. This implies that the total size of a sectoral flow only explains roughly one fifth to one quarter of the overall variability in uncertainties.

Thus, both approaches rely on simple, somewhat arbitrary assumptions to determine uncertainty estimates for the raw data. In the absence of better data on uncertainties, those assumptions might be justified. However, in the case of the data used to compile the GHG emission accounts of GMRIO, there indeed exists very detailed data on uncertainties, which to the best of our knowledge has never been applied to assess uncertainties of GMRIO extensions. One way to compile GHG emission accounts - referred to as inventory-first (Flachenecker et al., 2018) or top-down (Tukker et al., 2018) approach - is based on GHG

inventories from the United Nations Framework to Combat Climate Change (UNFCCC, covering only Annex-I countries) and/or EDGAR (covering more than 200 countries) (Crippa et al., 2020b). For both emission inventory data sources, UNFCCC and EDGAR, very detailed information on uncertainties is available. In case of the UNFCCC data, all parties submitting their national emission inventories to the UNFCCC are obliged to publish uncertainty estimates alongside them. However, those uncertainty estimates are hidden in the annexes of the so-called National Inventory Reports (NIR) which are only available in

PDF format. This makes them difficult to retrieve and process by computer, which in turn may explain why they have not been used to determine uncertainties of GMRIO databases so far. In case of EDGAR Solazzo et al. (2021) have recently estimated the uncertainties of the 2015 emission data.

      (2) The second shortcoming with respect to model uncertainty in GMRIO relates to the ignorance of correlations between variables introduced by disaggregating a common raw data item. Compiling an GMRIO is an undetermined problem, i.e. "there

are many more IO table entries than raw data items to construct them" (Lenzen et al., 2010). Therefore, raw input data items and their uncertainties have to be disaggregated. Lenzen et al. (2010) for example, apply a RAS-type balancing algorithm to "fit [...] an error propagation formula to the standard deviations of raw data" (Lenzen et al., 2013). By doing so, they ensure that the uncertainty of the disaggregate IO table entries is consistent with the "known" standard deviation of their common aggregate raw data item. Thereby they imply that the disaggregate uncertainties are uncorrelated and follow the standard error propagation

formula as formulated in Ku (1966). However, as Rodrigues (2016) showed, the assumptions of uncorrelated disaggregated variables on the one hand, and a known aggregate uncertainty on the other hand, are mutually exclusive. Rodrigues (2016) conclude that if "the aggregate uncertainty is known, prior correlations can be either all positive, all negative, or a mix of both, depending on the relative values of aggregate and disaggregate uncertainties." Ignoring correlations, in turn, might lead to an over- or underestimation of the model output uncertainty (Groen and Heijungs, 2017; Solazzo et al., 2021).

## 1.4 Goal and scope

Against this background, in this study, we aim to overcome the above listed limitations from previous approaches of assessing parametric uncertainty in GMRIO. First, we use available authoritative uncertainty estimates of the raw input data, instead of relying on simplistic assumptions. Thereby, we make use of uncertainty data from UNFCCC NIRs and Solazzo et al. (2021). Second, we include correlations, in particular those arising from data disaggregation, by sampling from Dirichlet distributions.





We estimate the uncertainty of the GHG emission accounts, thus leaving aside the uncertainty of the other two components of GMRIOs, i.e. the inter-industry matrix and final demand. GHG emission accounts are also referred to as GHG extensions or GHG satellite accounts (all three terms are used interchangeably in this work, also see tab. 1). We focus on the GHG emission accounts for three reasons: First, inter-database comparisons identified the GHG emission accounts as the major source of discrepancy between difference GMRIO databases (Owen et al., 2016), which makes them a relevant starting point for

improving the robustness of uncertainty estimates. Second, as detailed above, that is where we have authoritative information on raw data uncertainties so that we avoid "guestimating" them or basing them on (too) simplified assumptions. Third, as shown by Lenzen et al. (2010) and mentioned above, for carbon emissions the absolute size of a sector is an even poorer predictor of raw data uncertainty, than for financial transaction. Thus, we expect that the inclusion of more robust raw data uncertainties is especially relevant for GHG emission accounts.

We compile our own set of GHG emission accounts and estimate parametric uncertainty by using MC simulations to propagate the uncertainty from raw input data that enters the GHG extension compilation process to the GHG emission accounts, and then further to the GHG footprints.

We compile the GHG emission accounts for the year 2015 according to the sectoral and regional resolution of EXIOBASE v3 (Stadler et al., 2018) since it has the highest sectoral resolution of all currently available, harmonised (with respect to sector

resolution) GMRIO databases. We cover the three major GHGs $CO_2$, $CH_4$ and $N_2O$. As raw data for compiling GHG emission accounts, we follow recommendations by Tukker et al. (2018) and use, where available, emissions data from the UNFCCC as a "robust, authoritative source" (Tukker et al., 2018). We base the analysis on the Maximum Entropy Principle and thus try to use only the information that is available to use. Thus, we aim at providing a conservative base-line scenario of the uncertainty underlying the GHG emission accounts.

In addition to parametric uncertainty, we also provide an estimate of what Huijbregts (1998) call "uncertainty due to choices" reflecting the uncertainty that arises from decisions that inevitably have to be made in compiling and making use of GMRIO databases. Such decisions include for example the choice of the data sources used if there are several available (such as for GHG emissions), or the choice of how to allocate emissions to IO sectors. In GMRIO the uncertainty due to choices is either estimated specifically for one single decision using sensitivity analysis to study how the results differ when this decision

is made differently (Wiebe and Lenzen, 2016; Schulte et al., 2021), or "generally" by comparing the outcome of different GMRIO databases (Owen et al., 2016; Tukker et al., 2018). Those inter-database comparisons allow studying the variability in outcomes resulting from *all* decisions that have been made *differently* by the different database compilers in the course of compiling a GMRIO database. In this work, we choose the second approach by comparing our GHG emission accounts to those from other sources, namely the GHG emission accounts released by EXIOBASE v3.8.2, and official GHG emission accounts

published by national statistical agencies and collected by the OECD.

By doing so, we aim at guiding future GMRIO compilers to "uncertainty hot-spots" i.e. very uncertain data points which are relevant for answering a given research question. Such that resources can be more efficiently guided to improve data accuracy (if parametric uncertainty prevails) or improve (align) the overall compilation procedure (if uncertainty due to choices prevails).



The aim of the study is twofold: First, we estimate and present the uncertainty of both, GHG emission accounts (production-based perspective) and GHG footprints (consumption-based perspective), on two levels of detail - at the aggregate country-level and the disaggregate sector-level. Second, we provide GHG emission accounts along with their uncertainty to allow IO practitioners to conduct uncertainty assessment for their research question at hand.

## 2    Material and Methods

In the following, sec. 2.1 provides an overview of our methodology for compiling GHG emission accounts for EXIOBASE, including the raw input data and proxy data employed. Section 2.2 shows how we calculate GHG footprints using our GHG emission accounts along with data from EXIOBASE. Subsequently, sec. 2.3 outlines our approach to model the uncertainty related to those GHG emissions accounts, including the assignment of probability distributions to the input data (sec. 2.3.1) and the propagation of uncertainty of the input data through the compilation process to derive uncertainty estimates for the GHG emission accounts and the GHG footprints (sec. 2.3.2).

### 2.1    Compiling GHG emission accounts

#### 2.1.1    General information on GHG emission accounts

GHG emission accounts, also called GHG satellite accounts or GHG extensions (all three terms are used interchangeably in this work) represent GHG emissions broken down by emitting economic activity. Economic activities comprise both production and consumption activities. The System of Environmental-Economic Accounting (SEEA) provides the framework for the preparation of GHG emission accounts at the national level (UN et al., 2014). The SEEA frameworks shares the same system boundary as the purely economic System of National Accounts (SNA) to allow seamless integration between the economic Input-Output (IO) tables (based on SNA) and the environmental extensions. As such, the GHG emission accounts list all GHGs emitted within the *economic boundary* of an economic unit such as a country, thus following the *residential principle*. According to the residential principle, national GHG emission accounts list all emissions caused by residence units of a country. A residence unit is an institutional unit (e.g. a corporation, household, general government) which "has its centre of predominant economic interest in a particular economic territory" (UN et al., 2014). Emission accounts present GHG emissions from the production perspective. The design of the system boundary is one major difference between GHG emission accounts and other emission statistics such as national emission inventories reported to the United Nations Framework Convention on Climate Change (UNFCCC), which follow the *territorial principle* listing all GHGs emitted within the *geographical border* of a country (see Tab. 1).

Two approaches for constructing GHG emission accounts can be distinguished: the inventory-first and the energy-first approach (Eurostat, 2015). Both differ in the raw data used in the compilation process. While the inventory-first approach starts with the ready-made emission inventories, in the energy-first approach energy-accounts are constructed based on energy consumption data (such as the IEA World Energy Balances) and then combined with data on emission factors per fuel and economic





**Table 1.** Three different statistical perspectives on GHG emission.

| Statistical concept | Definition | Perspective | Synonyms | Abbreviation |
|---|---|---|---|---|
| GHG inventory | Emissions within geographical boundary | territorial | - | - |
| GHG emission accounts | Emissions within economic boundary | residential, production-based | satellite accounts, GHG extensions | GEA |
| GHG footprints | Emissions related to final consumption | consumption-based | consumption-based carbon accounts | CBCA |

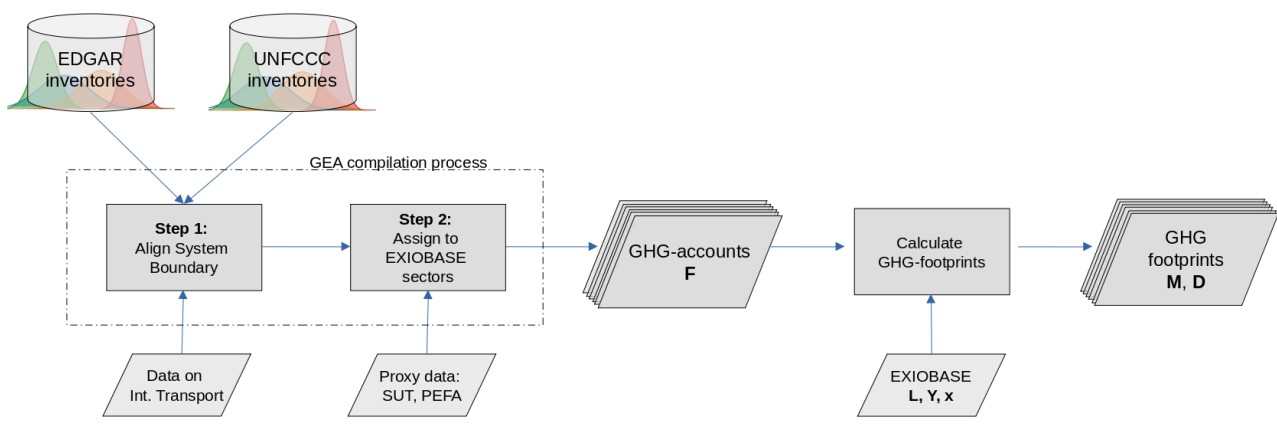

**Figure 1.** Our workflow of propagating uncertainty from the raw input data sources (UNFCCC/EDGAR) to the GHG emission accounts and further to the GHG footprints.

190 sector. The energy-first ensures database-internal consistency between energy and emission accounts (Stadler et al., 2018), but at the cost of a lack of consistency with emission data from authoritative sources such as the UNFCCC. Hence, In this study, we follow the recommendations from a recent review on the robustness of GMRIO (Tukker et al., 2018) and use the inventory-first approach. In appendix A, we provide a figure on the different data sources used to compile selected GHG emission accounts and how they differ (fig. A1).

195 Our workflow for the compilation of the GHG emission accounts and the calculation of GHG footprints is illustrated in fig. 1. We construct GHG emission accounts following the guidelines as described in Eurostat (2015). We use GHG emissions inventories as submitted to the UNFCCC and from EDGAR (Crippa et al., 2020b) as raw data input. We compile the GHG emission accounts according to the country and sector resolution of the industry-by-industry version of EXIOBASE v3, distinguishing 44 individual countries, 5 rest of the world (RoW) regions, each covering 163 industry sectors.

200 The GHG emission accounts compilation process can be divided into two major steps:

– Step 1: Aligning the system boundary to fit the residence principle





– Step 2: Assigning the emissions to EXIOBASE sectors

In the following, we first briefly describe the characteristics of the UNFCCC and EDGAR emission inventory data, and then outline the two steps in the GHG emission accounts compilation process.

### 2.1.2 Emission inventory data

We base our analysis on the GHG emissions inventories as submitted to the UNFCCC (hereafter referred to as UNFCCC data/inventories), and emission inventories from the Emissions Database for Global Atmospheric Research (EDGAR). Since 1994, all countries that agreed to take on greenhouse gas reduction commitments (the so-called Annex-I countries) have been required to report their territorial emissions on an annual basis as part of their commitment to the UNFCCC (UNFCCC, 1992). Currently, Annex-I countries (or parties) consist of 40 developed/industrialised countries and the EU. Reporting follows the IPCC 2006 guidelines (IPCC, 2006). The UNFCCC inventories as published by the UNFCCC secretariat were obtained from Pflüger and Gütschow (2020). For all non-Annex-I countries and few Annex-I countries for which we had problems while extracting uncertainty data (see sec. 2.3.1 and tab. C1 in the appendix), we use the emission inventories from EDGAR v5.0 (Crippa et al., 2020a) since they have a global coverage (231 countries plus bunker fuel emissions), they are published on a yearly basis, and, like the UNFCCC inventories, their reporting largely follows the IPCC 2006 guidelines. EDGAR data was obtained from Crippa et al. (2020b).

Both, UNFCCC and EDGAR classify GHG emission sources according to the Common Reporting Format (CRF). Whereas EDGAR uses the CRF as stated in IPCC (2006), the more recent UNFCCC inventories follow a slightly updated version (IPCC, 2019). In both CRF versions, emission sources are grouped into categories in a hierarchical order. In the case of UNFCCC inventories, the highest ranked categories - also called sectors - are 1) Energy, 2) Industrial Processes and Product Use, 3) Agriculture, 4) Land use, land-use change, and forestry (LULUCF), 5) Waste and 6) Other. Those sectors are further broken down into sub-categories, e.g. 1.A.1.a.i. The level of detail with regard to categories differs between the UNFCCC and EDGAR data, as well as between different countries. While EDGAR distinguishes only up to 22 different sub-categories, the national inventories submitted to the UNFCCC are much more granular, distinguishing up to 160 different sub-categories.

Furthermore, in case of the UNFCCC inventories the sub-categories of the sectors Energy, Agriculture (and LULUCF) are further broken down by the so-called classification. The classification distinguishes different fuel types (in case of emission from "Energy") and animal types (in case of emissions from "Agriculture"). Like the categories, the classification also follows a hierarchical structure, with a varying level of detail between individual country submissions. Hence, while for a certain category some countries only publish emission from "liquid fuels" or even only the "total for category", other countries distinguish e.g. "gasoline" and "diesel". EDGAR, on the other hand, does not provide details on fuel/animal type.

We exclude emissions from LULUCF as they are commonly not included in air emission accounts due to a lack of detailed enough data to allocate those emissions to industry and product sectors Eurostat (2015). Recently, efforts have been made to include LULUCF emissions into carbon footprint analysis (Hong et al., 2022), further research could include those in the uncertainty analysis framework presented here.

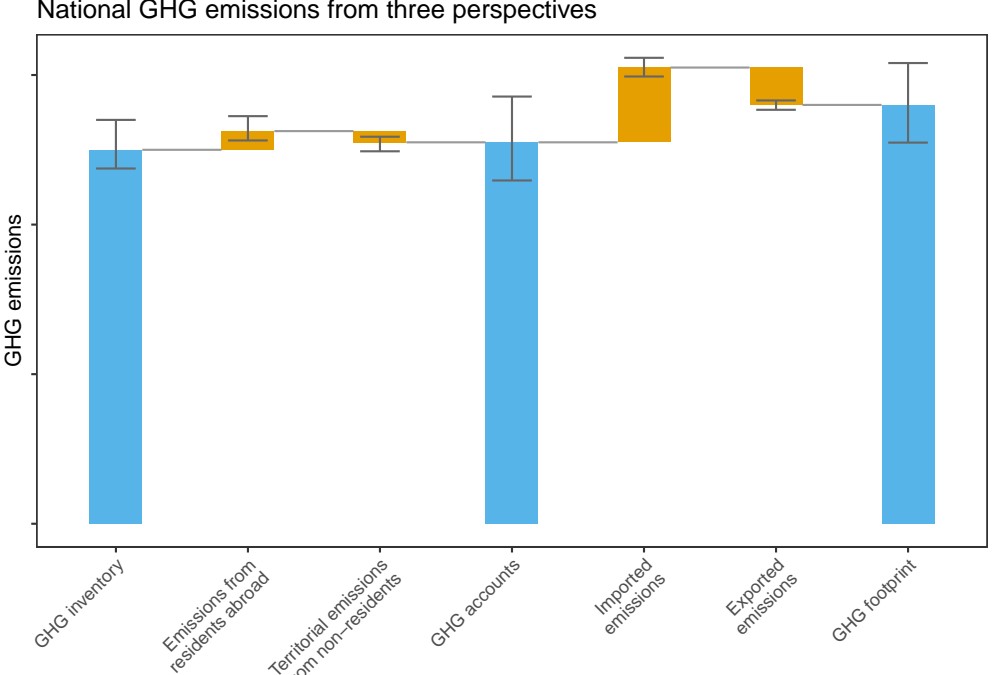

**Figure 2.** Schematic waterfall plot showing the relations between the three different statistical perspectives on GHG emissions. Error bars illustrate that each component is subject to uncertainty.

### 2.1.3   Step 1: Aligning the system boundary

As outlined above, emission inventories and emission accounts differ in their system boundary: While the former follows the territorial principle, the latter follows the residential principle. While a residence unit such as a company operates in most cases within the domestic territory, the exceptions can account for a considerable share of total national GHG emissions, especially for smaller economies like Luxembourg, Malta or Cyprus (Usubiaga and Acosta-Fernández, 2015) . Consequently, a crucial step in compiling GHG emission accounts is aligning the system boundary from the territorial to the residential principle, which is referred to as the 'residence adjustment'. (Eurostat, 2015). It should be noted, however, that not all global MRIO databases undertake this step (e.g. Eora (Lenzen et al., 2013) and ICIO (Wiebe and Yamano, 2016), see also fig. A1 in the SI), which was found to be a major reason of for the relatively large inter-database variability stemming from the GHG emission accounts (Owen et al., 2016).

The residence adjustment requires for each country/region (1) to deduct the emissions from non-resident units operating on the country's national territory, and (2) to add the emissions from resident units operating abroad (see Fig. 2). These operations are explicitly presented in the so-called 'bridging items' which cover both deduction and addition operations for all





relevant sectors. Thus, the net bridging items show the difference between national totals of territorial-based UNFCCC/EDGAR inventories and of national residence-based GHG emission accounts.

The residence adjustment affects a wide range of emission sources, however, due to the lack of data, estimating the bridging items is often hard. We therefore follow the compilers of EXIOBASE and WIOD and carry out the residence adjustment for the following globally relevant emission sources:

1. International air transport
2. International navigation
3. International fishing activities
4. International road transport

We note, that for some individual countries there might exist other quantitatively relevant items (e.g. pipeline transport), however, conducting individual residence adjustments for each country is beyond the scope of our paper.

Emissions from international air transport, international navigation (shipping) and fishing activities have in common that
they all occur, by definition, in international territory, namely in international airspace or in international waters. As emission inventories adhere to the territorial principle, these emissions are not included in national totals but rather classified as "memo-items". Adding up all memo-items from EDGAR delivers an estimate of the global total emissions from international air and maritime activities, respectively. Consequently, these memo-items need to be added to the corresponding EXIOBASE sectors, namely "Air transport", "Sea and coastal water transport" and "Fishing, operating of fish hatcheries and fish farms; service
activities incidental to fishing" of the countries that are home to the emitting institutional units. All emissions caused by the Irish airline Ryanair, for example, need to be added to the Irish air transport sector, because this is the sector where the purchases of the airline's customers will be recorded in the IO tables.

To insure consistency with the total global emissions from international air and maritime activities from EDGAR, we calculate country specific use shares for both activities using additional auxiliary data (details in appendix D). Those use shares
(which globally sum to one) are multiplied with the total global international air/maritime emissions from EDGAR. In case of international maritime activities, we calculate country specific use shares using data from Selin et al. (2021). In their analysis, Selin et al. (2021) made a bottom-up estimate for the allocation of $CO_2$ emission from international shipping to national carbon budgets for the year 2015 using spatially-resolved data on ship movements and ship-specific data on engine power demand, activity time and emissions factor (details in appendix D1). In case of international air transport, we calculate country spe-
cific use shares for EU countries using bridging items provided by Eurostat (Eurostat, 2022), and for non-EU countries using data from the World Bank on the country-specific numbers of domestic and international air passengers carried by air carriers registered in the country (Worldbank, 2023). For more details, see appendix D2.

Emissions from international road transport occur within national territory but are caused by non-residence. As such international road transport includes both tourism activities and road freight transport. These emissions need to be added to the
corresponding EXIOBASE sector of the residence country of the emitter. For instance, emissions resulting from a tourist driving a car abroad would be allocated to the household sector of their home country. Similarly, emissions caused by a logistics





company operating abroad would be added to the "Other land transport" sector of the country where the logistic company is registered. Moreover, as opposed to emissions from international territory, emissions from international road transport additionally need to be subtracted from the account of the country where the emissions occur (or more precisely from the country

where the fuel was sold since emission inventories mostly use fuel sale statistics to estimate transport emissions).

In case of international road transport, we follow Stadler et al. (2018) and consider the European countries to be by far the most affected by the residence adjustment of international road transport due to the European geography and its economic size (many countries in relatively small area with a lot of cross-border commercial and recreational road transport). Non-European regions represented in EXIOBASE are "either islands or countries with limited road access in relation to their size

(e.g. China, India)" (Stadler et al., 2018). We therefore assume that for those countries the road transport emissions from non-resident units operating on the country's national territory, and the emissions from resident units operating abroad, are the same. In other words, we assume the bridging items related to international road transport for non-EU countries to be zero. However, in contrast to international air and water transport we have no knowledge on the total (global/European) emissions from international road transport, as they are not reported separately as memo items but as part of the road transport sector

emissions (CRF category 1.A.3.b) within each country. That is why - instead of calculating use shares as we do for water transport - we directly take the total bridging items from Eurostat (Eurostat, 2022) and add/subtract those from the respective EU-country's national road transport emissions. Eurostat's bridging items do not sum to zero, thus we distribute the residual to all European countries not listed in the Eurostat data using the total emissions from road transport (1.A.3.b) as country-specific proxy. For details, please refer to the appendix D2.

**2.1.4 Step 2: Assigning emissions to MRIO sectors**

Next to the differences in the system boundary, emission inventories and emission accounts also differ in their classification scheme. While inventories have more technical process-oriented classifications, emission accounts are grouped according to economic activity. Using the example of road transportation: In the UNFCCC CRF, emissions from road transportation are broken down according to emitting vehicle: Cars, Light Duty Trucks, Heavy Duty Trucks, Motorcycles, and Others. Thereby,

focusing on differences in technology while ignoring the institutional unit that operates the vehicle, i.e., whether the operator is a household or a transport company. In contrast, emission accounts would allocate these emissions from road transportation to the operators of the vehicles, thus to households, the logistics sector, and all economic sectors that operate vehicles (which are basically all economic sectors, details see appendix D3).

Creating a correspondence table (CT) is the first step for assigning the inventory emission sources to EXIOBASE sectors.

The CT needs to map each combination of CRF category and classification (in the following referred to as "CRF emission source", e.g. "Liquid fuel" emission from category "1.A.1.a"), for *each level of detail*, to the EXIOBASE sectors differentiating individual industry sectors and final demand categories. To get a CT that is consistent over all hierarchical levels, we manually constructed the CT for the most detailed combinations among all countries (which we name root classification, following Lenzen et al. (2013) ). The upper level is then filled automatically by merging the respective EXIOBASE correspondences

from the lower levels.





As a starting point, we take the correspondence table published by Eurostat that maps UNFCCC categories (without classification detail) to NACE rev2 sectors (until level 2, e.g. C11). Since we aim for a higher level of detail at both sides of the CT, a considerable effort was needed to create the CT. We follow recommendations from Eurostat (2015) for creating correspondence tables by taking the following steps: First, we get a detailed understanding of the CRF categories and classifications (fuel and animal types) based on the IPCC 2006 guidelines (IPCC, 2006); EMEP and EEA (2019), and National Inventory Reports (NIR). Second, we get a detailed understanding of EXIOBASE sectors using the official documentations (Stadler et al., 2018, and supplementary material) and, where more detail is needed, the documentation of the NACE rev2 classification (on which the EXIOBASE classification builds upon). Third, we assign corresponding EXIOBASE sectors to each combination of CRF category and classification from the root classification. We end up with both, 1-to-1 correspondences and 1-to-N (many) correspondences. Moreover, some CRF category/classifications which are found to be *country specific*, are marked and in a first step the correspondence is chosen according to the German inventory.[1] An upcoming work will be to construct *country-specific* CTs (or even country-year-specific, since correspondences within a country might also change over time). Since CRF emission sources marked as country-specific make up only a small share of total emission from Annex-I countries (0.4% for $CO_2$, 5.7% for $CH_4$, 1.2% for $N_2O$), we consider our results not be significantly affected by this pragmatic choice. Fourth, for each 1-to-N correspondence, we identify suitable proxy data to get a best-guess estimate for allocating (disaggregating) the CRF emission sources to MRIO sectors. As proxy data sources, we use in most cases the (monetary) Supply and Use Tables (SUT) from EXIOBASE. To split emissions from Road Transport (CRF category 1.A.3.b) to EXIOBASE sectors we use Physical Energy Flow Accounts (PEFA) (Eurostat, 2023) along with the industry sector specific employment data from the EXIOBASE extensions (Stadler et al., 2018) as proxy data. Our CT including all UNFCCC-EXIOBASE mappings and their proxy data sources is available on Zenodo (see data availability at the end of the manuscript).

**Special case: Road transport** Since road transport activities are undertaken by basically all industries and households, the allocation of emissions from road transport (CRF category 1.A.3.b) is one of the most difficult parts of compiling GHG emission accounts (Eurostat, 2015). Since the availability and quality of auxiliary data needed to estimate the shares of the respective industries and households in total road transport emission is highly country specific (Eurostat, 2015), we use in a first step Eurostat's Physical Energy Flow Accounts (PEFA) to allocate road transport emissions to NACE rev2 industries and households for all EU28 countries plus Iceland and Norway (Eurostat, 2023). Given that the PEFA solely incorporates NACE rev2 industries up to the 2nd level (e.g. C11), further disaggregation is needed to align with the more detailed sector classification of EXIOBASE. Consequently, in a second step, we further disaggregate emissions from NACE-sectors to EXIOBASE sectors using sector specific employment data from the EXIOBASE extensions (Stadler et al., 2018), more specifically, the sum of the working hours from low-, middle- and high-skilled employment. In our analysis, we deem the total working hour input to be a more suitable proxy for allocating road transport emissions compared to the economic output of a sector. This choice is predicated on the assumption that the number of business trips (a major source of industry road transport emissions) is

---

[1]Germany was chosen for two reasons: First, it is a major GHG emitting country, and second, with ± 1000 pages the German NIR is one of the most detailed reports of all submitting countries.





primarily contingent on the workforce size within a sector, rather than solely relying on its overall output. For a more detailed and formal elaboration on how we allocate road transport, we refer the reader to the appendix D3.

## 2.2 Calculate GHG footprints

We transform the GHG emission accounts compiled with the procedure detailed above into a matrix $\mathbf{F}^*$. The columns of $\mathbf{F}^*$ represent the 7987 industry-country combinations following the structure of EXIOBASE v3, while the rows represent the 33 combinations of the three GHGs ($CO_2$, $CH_4$, $N_2O$) and 11 emission sources according to the UNFCCC CRF. By combining our GHG emissions accounts $\mathbf{F}^*$ with the other elements needed from EXIOBASE v3.8.2 (Stadler et al., 2021, 2018), namely the inter-industry coefficient matrix $\mathbf{A}$, the sectoral output $\mathbf{x}$ and the final demand matrix $\mathbf{Y}$, we first calculate the matrix of environmental multipliers $\mathbf{M}$ storing the consumption-based environmental impacts to produce one unit of output by industry sector:

$$\mathbf{M} = \mathbf{F}^* \hat{\mathbf{X}}^{-1} (\mathbf{I} - \mathbf{A})^{-1} \,, \tag{1}$$

where $\mathbf{I}$ is the identity matrix and $\hat{\mathbf{X}}^{-1}$ is a square matrix with $1/x_i$ on the main diagonal and zeros elsewhere.

We calculate national footprints $\mathbf{D}$ as

$$\mathbf{D} = \mathbf{M}\mathbf{Y}. \tag{2}$$

## 2.3 Uncertainty analysis

We use Monte-Carlo (MC) simulations to propagate uncertainty from the raw data (GHG inventories from UNFCCC and EDGAR) to the GHG emission accounts and then further to the GHG footprints. Uncertainty propagation using MC requires first to assign probability distributions to the raw input data. Subsequently, we perform MC simulations by repeatedly ($N = 1000$) and randomly sampling from those probability distributions. We use those $N$ random samples to create a set of $N$ GHG extension matrices $\{\mathbf{F}_1^*, \mathbf{F}_2^*, ..., \mathbf{F}_N^*\}$ following the procedure described sec. 2.1. Which in turn are then used to calculate sets of $N$ multiplier $\{\mathbf{M}_1, \mathbf{M}_2, ..., \mathbf{M}_N\}$ and of $N$ national footprints $\{\mathbf{D}_1, \mathbf{D}_2, ..., \mathbf{D}_N\}$ (sec. 2.2). In the following, we first show how we handle data uncertainty of the raw data, and then explain in detail how we model uncertainty propagation.

### 2.3.1 Assigning probability distributions to input data

Uncertainties of national GHG emissions inventories as submitted to the UNFCCC are available in the National Inventory Reports (NIR). NIRs are published annually by Annex-I countries along with the actual emission data (see sec. 2.1.2). The reporting of the uncertainties in the NIRs largely adheres to the IPCC 2006 guidelines (IPCC, 2006), specifically the template table for uncertainty reporting found in Tables 3.2 and 3.3 of Volume 1, Chapter 3. Since the NIRs are only available in pdf-format we first had to extract the uncertainty tables using set of python scripts. The extracted uncertainty estimates from the 2017 submission of UNFCCC NIRs covering the year 2015 are available on Zenodo (see data availability at the end of



the manuscript). Note, that we were not successful in extracting uncertainty data for all Annex-I countries due to the lack of uncertainty data in those countries' NIRs, or other issues which inhibited extracting or processing those data. In tab. C1 we list all EXIOBASE countries and regions, along with the database we used as raw data source.

The NIR uncertainty tables list the uncertainties by source category (e.g. 1.A.3) and classification (i.e. fuel or animal type). The uncertainties are either given as one value representing a symmetric 95% confidence interval (CI) around the mean (two relative standard deviations: $2\sigma$), or a lower and upper uncertainty bound which enclose the 95% CI in the form of $(Q_{0.025}, Q_{0.975})$, where $Q_{0.025}$ and $Q_{0.975}$ are the 2.5th and 97.5th percentiles, receptively. The type of uncertainty reporting depends on whether the reporting country estimated the emission uncertainties based on analytical error propagation where

uncertainty in emissions is propagated from uncertainty of the activity data, emission factors and other parameters using the error propagation equation (Ku, 1966) (approach 1), or based on Monte-Carlo simulations (approach 2, see IPCC (2006)).

For EDGAR data which we use for all non-Annex I countries and those Annex-I countries for which we could not extract the data from the NIRs, uncertainties are available from Solazzo et al. (2021). Solazzo et al. (2021) apply a similar approach by also following the 2006 IPCC guidelines (IPCC, 2006). Compared to the uncertainties reported by the UNFCCC, however, they

mostly use default emission factor uncertainties from IPCC (2006) thus omitting national peculiarities. Like the uncertainty data from NIRs, Solazzo et al. (2021) report the EDGAR uncertainties either as symmetric or asymmetric 95% CIs.

In case of symmetric (approach 1) uncertainties, we assign a truncated normal distribution $Truncnorm(\mu, \sigma, a = 0)$, where $\mu$ is the mean taken from the UNFCCC/EDGAR inventory data, $\sigma$ is the standard deviation taken from UNFCCC NIRs or Solazzo et al. (2021), respectively, and $a$ depicts the minimum value.

In case of asymmetric (approach 2) uncertainties, we assign a log-normal distribution $Lognormal(\mu^*, \sigma^*)$, where $\mu^*$ and $\sigma^*$ are the mean and standard deviation of the variable's natural logarithm. $\mu^*$ and $\sigma^*$ are both estimated so that the 95% CI of the log-normal distribution fits the 95% CI as given in UNFCCC NIRs or Solazzo et al. (2021) using the R-package "rriskDistributions" (Belgorodski et al., 2017).

One major challenge for both emission inventories (UNFCCC and EDGAR) is that the level of detail of the uncertainty data often does not match the level of detail of the emission data. This mismatch in resolution is present in both categories and

classifications (i.e. fuel/animal types, applying to UNFCCC only since EDGAR does not distinguish fuel/animal types).

Figure 3 exemplifies this mismatch for the CRF-category 1.A.2 comprising emissions from fuel combustion from 'Manufacturing Industries and Construction' and its sub-categories. In that example, we have emissions data up to the fourth category level (1.A.2.a Iron and Steel, 1.A.2.b Non-Ferrous Metals, etc.) each for three different fuel types (see the circles outlined in

light blue). Uncertainty data, however, is only available at the third category level (1.A.2) without any details on fuel type (see the circles filled in orange). The easy solution to deal with this mismatch in granularity would be to use the data at the level of detail for which both emissions and uncertainty data is available. This option, however, would come at the cost of losing valuable information on the composition of the emission sources, so that we would have to make even more assumption regarding the allocation of emissions to MRIO sectors. As such, in order to use all information available, we handle the data

in a hierarchical tree format, in two different variants either based on (A) the category, or (B) the classification. So that we have one data tree for each party, year, gas, and *classification* (in case of A), or one data tree for each party, year, gas, and

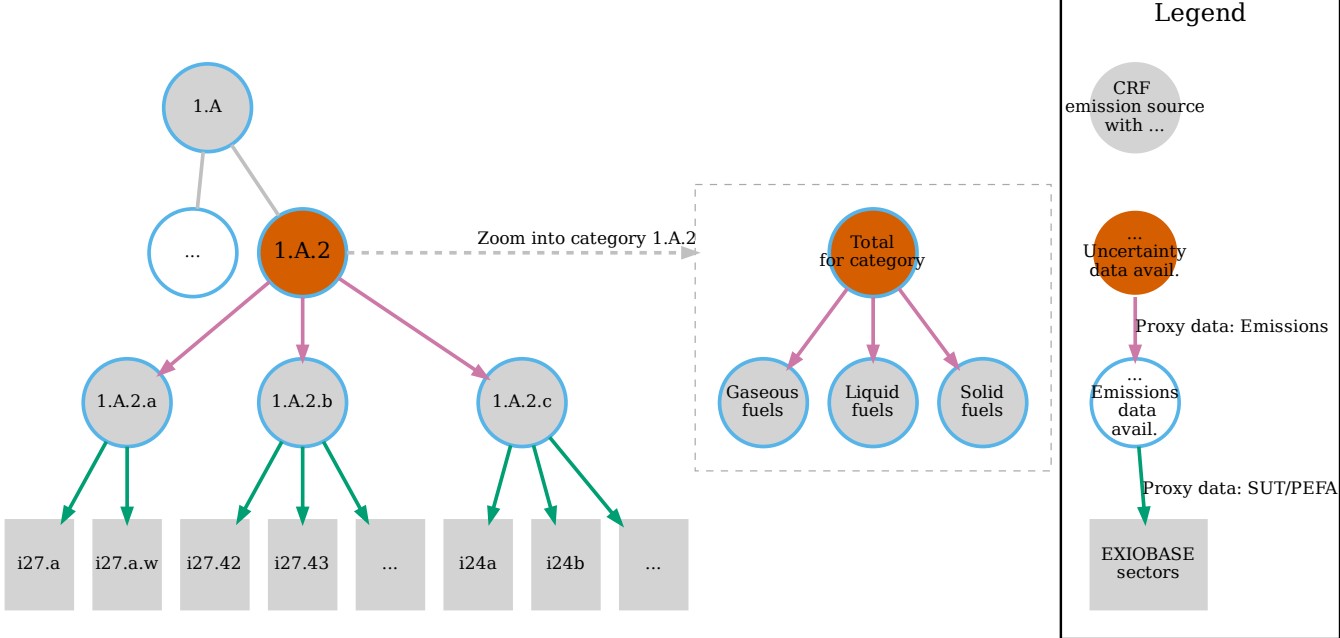

**Figure 3.** An example of the nested hierarchical data structure of the UNFCCC inventories: One hierarchy representing the categories (left). Each node representing a category contains another hierarchy representing the classification (fuel/animal type).

*category* (in case of B), respectively. We wrote functions to flexibly reshape the data between the usual table format and the category/classification tree formats.

The modelling of uncertainty of the residence adjustment (sec. 2.1.3) differs between international road transport on the one side, and international air and water transport on the other side. While in case of the former we have no information on the total (global) emissions from international road transport but only the country-specific bridging items, in case of the latter we know the total (global) emission from international air and water transport from the UNFCCC and EDGAR memo items (see sec. 2.1.3) along with their uncertainties (see next chapter).

That is why, for international road transport, we explicitly need to assign uncertainty estimates to the country-specific bridging items, while for international air and water transport we can model the uncertainty of the country-specific bridging items by disaggregating the global international air/navigation emissions using the procedure presented below. Since Eurostat does not provide uncertainties of their bridging items, we assume a relative standard deviation of 0.3. Note that due to taking totals (instead of use shares) also the uncertainty propagation differs between the residence adjustment related to international road transport as opposed to emission happening in international territory.

### 2.3.2 Propagating uncertainty

We propagate the uncertainty using 1000 MC simulations in three steps:





1. from the nodes that have uncertainty information (fig. 3, orange circles) to the most detailed level to which emissions data is available (fig. 3, lowest level blue bordered circles),

2. from these inventory leaves further to the MRIO sectors (extensions), and

3. from the GHG emission accounts to the GHG footprints.

The aforementioned fig. 3 illustrates steps (1) and (2) of the uncertainty propagation procedure. Steps (1) and (2) involve both 1-to-1 mappings (i.e. node is only connected to *one* lower-level node, not shown in fig. 3 and 1-to-N mappings (i.e. node is connected to two or more lower-level nodes). While the first case is trivial, the second one involving data disaggregation requires further attention.

The problem of data disaggregation under uncertainty (i.e. in a probabilistic framework) appears in many different research fields (e.g. chemistry: Plessis et al. (2010), economics: Rodrigues (2014), energy: Paoli et al. (2018), Min and Rao (2018)). The main characteristic of the problem of data disaggregation is the preservation of the *accounting identity*, i.e. the constraint that all disaggregate data values $x_1, ..., x_K$ need to sum to the aggregate data value $x_0$

$$x_0 = \sum_{i=1}^{K} x_i \tag{3}$$

Rewriting Equation (3) by substituting $\frac{x_i}{x_0} = \alpha_i$ gives:

$$x_0 = x_0 \sum_{i=1}^{K} \alpha_i, \tag{4}$$

where $\alpha_i$ is the branching ratio (or sector share) of sector $i$, and $\sum_{i=1}^{K} \alpha_i = 1$. The accounting identity constraint naturally introduces negative correlations between the $\alpha_i$'s (Rodrigues, 2016).

We approach the problem of data disaggregation under uncertainty as follows: First, we sample the aggregate $x_0'$ from the

uncertainty distribution for $x_0$ assigned in section 2.3.1. Second, we sample the disaggregate branching ratios $\boldsymbol{\alpha}' = \alpha_1', ..., \alpha_K'$ from the Dirichlet distribution of $\boldsymbol{\alpha} = \alpha_1, ..., \alpha_K$ which will be detailed in Schulte et al. (in preparation). Together, $x_0'$ and $\alpha'$ then provide the sampled disaggregate values: $x_i' = x_0' * \alpha_i'$.

For data disaggregation in a probabilistic framework, the Dirichlet distribution is often a natural choice (see Paoli et al., 2018, e.g) since it has the helpful properties that random variables drawn from the distribution always sum to 1. Formally expressed,

the Dirichlet distribution describes $K \geq 2$ random variables $X_1, ..., X_K$ such that each $x_i \in (0, 1)$ and $\sum_{i=1}^{K} x_i = 1$. The Dirichlet distribution we use which is described in Plessis et al. (2010) is parameterised as follows:

$$x_1, ..., x_K \sim Dir(\alpha_1, ..., \alpha_K; \gamma), \tag{5}$$

where $\boldsymbol{\alpha} = (\alpha_1, ..., \alpha_K)$ is a vector of positive-valued parameters such that $\sum_{i=1}^{K} \alpha_i = 1$, and an additional positive-valued concentration parameter $\gamma > 0$. The Dirichlet distribution, as described here, has the useful property that the expected values

for each variable $X_i$ equal the parameter value $\alpha_i$:





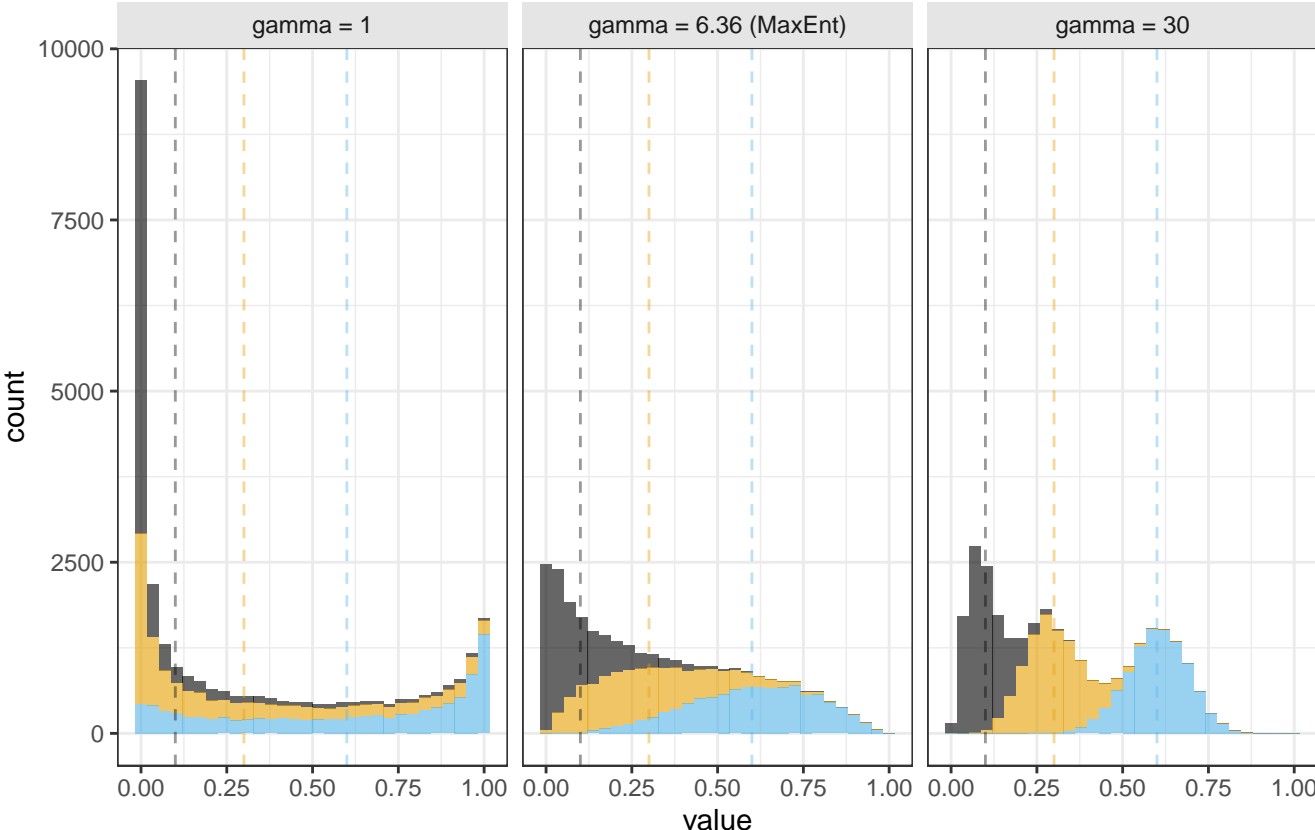

**Figure 4.** Histograms of sector shares for three sectors (grey, yellow, blue) sampled from Dirichlet distributions with different values of gamma (N = 10000, $\alpha = (0.1, 0.3, 0.6)$, $\gamma \in \{1, 6.36, 30\}$)

$$E[X_i] = \alpha_i, \forall i \in \{1, ..., K\}. \tag{6}$$

The concentration parameter $\gamma$, on the other hand, controls the variance of $X$. This is illustrated in fig. 4 showing histograms of 10000 random numbers generated with three different Dirichlet distributions, all with the same average sector shares $\boldsymbol{\alpha} = (0.1, 0.3, 0.6)$, but with different values of $\gamma$. From the figure, we can see that the variance decreases with increasing $\gamma$.

In other words, with the parameter $\gamma$ we can introduce uncertainties of the sector shares. However, we have no information on how accurate the SUT and PEFA data is as a proxy for disaggregating emissions data. Thus, without quantitative information uncertainties of the sectors shares, we cannot choose one of these realisations of the Dirichlet distribution without explicitly making an (arbitrary) assumption on the uncertainty (i.e. variance) of the shares. Against this background, the Maximum Entropy (MaxEnt) principle provides a powerful framework to deal with all available information and constraints in a consistent

manner. According to Jaynes (1957), from all probability distributions that align with a given set of constraints and information, the one with the maximum entropy should be selected. The MaxEnt principle implies that the chosen distribution is at the





same time maximal uninformative about what is unknown, and maximal informative about what is known. Consequently, the MaxEnt distribution provides the least biased estimation that remains consistent with the provided constraints and information. Thus, in our case we want to find the least informative - or least biased - Dirichlet distribution with given sector shares $\boldsymbol{\alpha}$.

Or more precisely, we estimate the concentration parameter $\gamma$ such that the entropy of the Dirichlet distribution $Dir2(\boldsymbol{\alpha}; \gamma)$ is maximised. A more detailed elaboration of our procedure is under preparation (Schulte et al., in preparation).

## 3 Results

Here, we present the results from propagating uncertainties from the UNFCCC and EDGAR emission inventories to the GHG emission accounts, and further to the GHG footprints. The results section is structured as follows. First, section 3.1 shows

the uncertainty of the GHG emission accounts and the GHG footprints at the level of countries/regions, and compares our range estimates to the point estimates from the official EXIOBASE GHG emission accounts and - if available - GHG emission accounts published by national statistical offices (and collected by the OECD). Subsequently, section 3.2 shows the uncertainty at the level of industry sectors.

We provide 95% confidence intervals (CI) $[Q_{0.025}, Q_{0.975}]$ depicting the interval between the 2.5$^{th}$ and 97.5$^{th}$ percentiles of

the 1000 Monte-Carlo samples.

### 3.1 Uncertainty at the regional/country level

Figure 5 displays the uncertainty of the GHG emission accounts at the country level for the three GHGs $CO_2$, $CH_4$ and $N_2O$. The grey bars show the uncertainty as the relative deviation of the 95% CI from the sample mean. For $CO_2$, the uncertainties range between minimally [-2%, +3%] for WM, and maximally [-35%, +257%] for MT. For $CH_4$, the uncertainties range

between minimally [-8%, +10%] for WE, and maximally [-50%, +59%] for BG. For $N_2O$, the uncertainties range between minimally [-26%, +34%] for WM, and maximally [-48%, +203%] for MT. Thus, the uncertainty ranges span almost factor 100 for $CO_2$, and less than factor 10 for $N_2O$ and $CH_4$. Moreover, for most countries, the 95% CI is positively skewed. This can be explained by the constraint that emissions can only be positive, thus the theoretically maximum relative downward deviation from the mean is -100%.

Comparing our range estimates to the point estimates from the official EXIOBASE GHG emission accounts and the collection of national emission accounts from the OECD (coloured points in fig. 5, summary statistics in fig. E1 in the appendix), we observe that for most countries, both, EXIOBASE and OECD estimates fall within our 95% CI. Exceptions include some countries (AT, AU, BE, EE, FI, GB, HU, PT, SE) for which the $CH_4$ estimate from EXIOBASE is well above our 95%, in case of Finland even by factor more than 3. We suspect that those discrepancies can mostly be explained by different source data

used, which for those countries also differ considerably. While EXIOBASE estimates align well with the EDGAR inventory data, our estimates and OECD estimates both are based on the UNFCCC inventories. Therefore, we conclude that for those countries, the uncertainty due to choices (in this case the choice of database) is much higher than the parametric uncertainty.



**Figure 5.** Uncertainty of the GHG emission accounts for the 49 EXIOBASE countries and regions, shown as the relative deviation from the sample mean. Grey bars show our 95% CI. The blue and yellow points show the relative deviation of the official EXIOBASE v3.8.2 emission accounts and the collection of national emission accounts from the OECD, respectively, from our sample mean. EDGAR and UNFCCC emission inventories (territorial-based) are additionally displayed to help explaining some of the differences between our GHG emission accounts and those from the OECD or EXIOBASE. See appendix C1 for a definition of all EXIOBASE country/region codes.





**Table 2.** Distribution of the coefficients of variation (CV) of the country- and sector-level GHG emission accounts and GHG footprints. Numbers are denoted in the form of $median^{+(Q_{0.975}-median)}_{-(median-Q_{0.025})}$

| level | gas | GHG emission accounts | GHG footprints |
|---|---|---|---|
| country/region | CH4 | $0.12^{+0.13}_{-0.07}$ | $0.06^{+0.1}_{-0.03}$ |
| country/region | CO2 | $0.04^{+0.46}_{-0.02}$ | $0.03^{+0.13}_{-0.02}$ |
| country/region | N2O | $0.33^{+0.25}_{-0.17}$ | $0.16^{+0.28}_{-0.08}$ |
| economic sector | CH4 | $1^{+21.63}_{-0.81}$ | $0.1^{+1.11}_{-0.06}$ |
| economic sector | CO2 | $0.94^{+20.76}_{-0.82}$ | $0.18^{+3.57}_{-0.15}$ |
| economic sector | N2O | $1.13^{+20.89}_{-0.88}$ | $0.22^{+1.58}_{-0.16}$ |

Also in case of the four EXIOBASE Rest of the World (RoW) regions, EXIOBASE estimates mostly fall outside our 95%. This might result from the fact that in contrast to most countries that are individually present in EXIOBASE, there is no official

"benchmark" estimate for the emissions of those RoW regions. In case of the $CO_2$ emission accounts from CH and SE, both EXIOBASE and OECD estimates are below our 95% CI. In case of $N_2O$, we observe that for most countries the EXIOBASE and OECD estimates are systematically below our sample mean (but still within our 95%). An exception to that is Australia, where EXIOBASE reports 250% higher (residential-based) $N_2O$ than our sample mean. But, similar to the $CH_4$ 'outlier', we also suspect the different source data to be (one) explanation for this discrepancy.

Figure 6 displays the uncertainties of the GHG emission accounts next to the uncertainties of the GHG footprints, allowing to analyse how the uncertainty propagates from the production-based GHG emission accounts through international supply chains to the consumption-based GHG footprints. This time, the countries are sorted along the x-axis according to their mean share of total emissions (according to our estimate) - from low (left) to high (right).

Focusing on the uncertainty of the GHG emission accounts (6, left column), we see that in the case of $CH_4$ and $N_2O$, there

is no clear trend between a country's emissions' uncertainty and its absolute emissions. This means that the uncertainty is relatively uniform between countries, regardless of the size of their total emissions. In contrast, for $CO_2$, a clear trend emerges where countries with larger overall GHG emission accounts exhibit lower uncertainty. Those countries with by far the greatest uncertainty (Malta, Cyprus, and Luxembourg) all have very small (production-based) contributions to global $CO_2$-emissions. Countries and regions with a considerable contribution to global $CO_2$-emissions, on the other hand, such as China, the US,

RoW Middle East, or India show relatively small uncertainties with CIs all ranging within the interval [-10%,+10%].

Comparing the uncertainty of the GHG emission accounts (production-based) to the uncertainty of the GHG footprints (consumption-based), we observe that for most countries the uncertainty of the former is considerably higher than of the latter. This can also be seen in tab. 2, which shows the distributions of the coefficients of variations (CV) as the median (50th percentile) and the 2. and 97.5 percentiles.

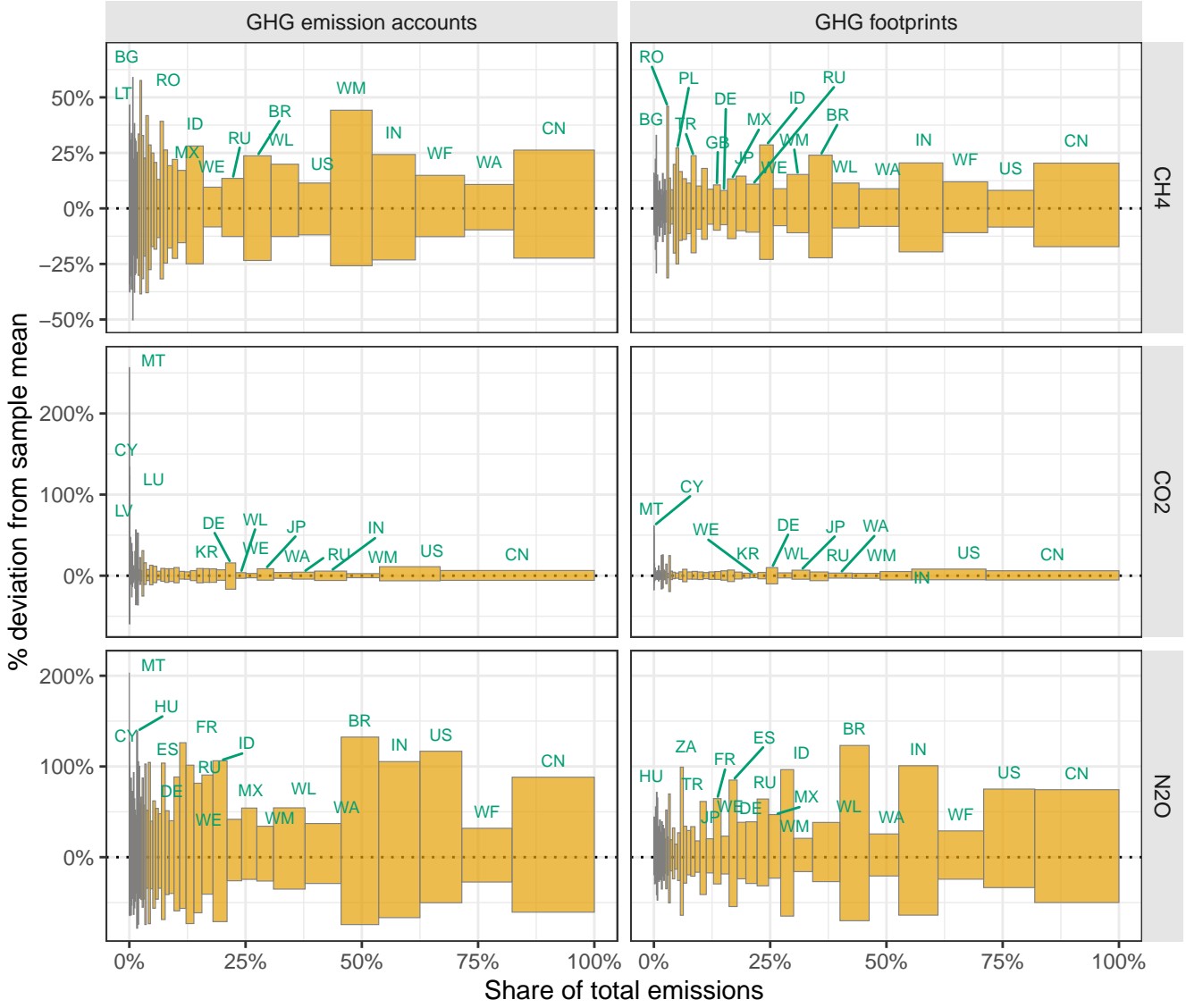

**Figure 6.** Uncertainty of the GHG emission accounts (left) and of the GHG footprints (right) at the country/region level, shown as the relative deviation from the sample mean. Yellow bars show our 95% CI. Countries are sorted along the x-axis according to their mean share of total emissions. The bar width is adjusted to the mean share of total emissions. For EXIOBASE country/region codes see C1.



This difference is particularly striking in countries with very uncertain GHG emission accounts, such as Malta and Cyprus, especially concerning $CO_2$ and $N_2O$ emissions. The lower uncertainty in GHG footprints compared to GHG emission accounts can be attributed to the fact that when calculating footprints, the emissions from GHG emission accounts - and their associated uncertainties - are distributed internationally through global supply chains. For example, a large share of Malta's 'sea and coastal water transport' services, whose emissions are a major source of uncertainty (see fig. E2 in the appendix), is not

consumed domestically, but relates to final consumption in other parts of the world. However, it must be noted that in our analysis we only consider the uncertainty of the GHG emission accounts, thus neglecting the uncertainty stemming from all inter-industry flows including how emissions are internationally distributed through global supply chains. Hence, depending on the size of those uncertainties not covered here, the ratio might also be reversed.

### 3.2   Uncertainty at the sectoral/multiplier level

Next, we turn to the uncertainty at the level of industry sectors. The industry-by-industry version of EXIOBASE v3.8.2 covers 163 economic sectors in each of 49 countries and regions, making a total of 7987 economic sectors. We analyse both the uncertainty of the sectoral GHG emission accounts, and the uncertainty of the sectoral GHG footprints. In case of sectoral GHG footprints, the consumption-based emissions to produce one unit of output is also often referred to as (emission) multiplier.

    Figure 7 displays the uncertainty relative to the mean in the (production-based) GHG emission accounts of those economic

sectors. Each economic sector is represented by one grey bar showing the range of its 95% CI (y-axis) and its share in total (global) emissions (x-axis). The sectors are sorted along the x-axis according to their mean share of total emissions. The coloured points depict the EXIOBASE v3.8.2 estimate for that sector. Colours indicate if the EXIOBASE estimate is above, below or within our 95% CI.

    The right column of fig. 7 shows all sectors (except those with zero emissions). From there we can see that for some

sectors with a very small contribution to total emissions, the EXIOBASE estimates deviate substantially by up to factor +1E9. However, since those sectors only make a very small contribution to global emissions but completely dominate the scale of the y-axis, we zoom into those sectors which cover the top 80% of total emissions (left column of fig. 7). For those top-80% sectors, the 95% CI ranges between [-100%, +200%] for $CH_4$, [-100%, 300%] for $CO_2$, and [-100%,350%] for $N_2O$.

    For $CO_2$, similar to the country-level uncertainties, there can be observed a clear trend between a sector's emissions' uncer-

tainty and its absolute emissions. For example, the 95% CIs of the sectors covering the top 40% of total $CO_2$-emissions are all within [-50%,+50%], while for $N_2O$ the sector with the highest share of total emissions (China - Cultivation of vegetables, fruit, nuts) shows a 95% CI of [-85%, 140%], and for $CH_4$ the sectors with the third-highest share of total emissions (China - Mining of coal and lignite; extraction of peat) shows a 95% CI of [-70%,+90%]. Moreover, like for country-level uncertainties, for most sectors the 95% CI is positively skewed.

Comparing our range estimates (covering the 95% CI) with the point estimates from EXIOBASE we see - in contrast to the country-level results - a considerable deviation between the EXIOBASE estimates and our 95% CIs (coloured points in fig. 7, fig. E1 in the appendix). All sectors for which the EXIOBASE estimate falls within our 95% CI make up 28% ($CO_2$), 35% ($CH_4$) and 41% ($N_2O$), respectively, of global emissions (see fig. E1 in SI). For most sectors, making up 58% ($CO_2$), 47%

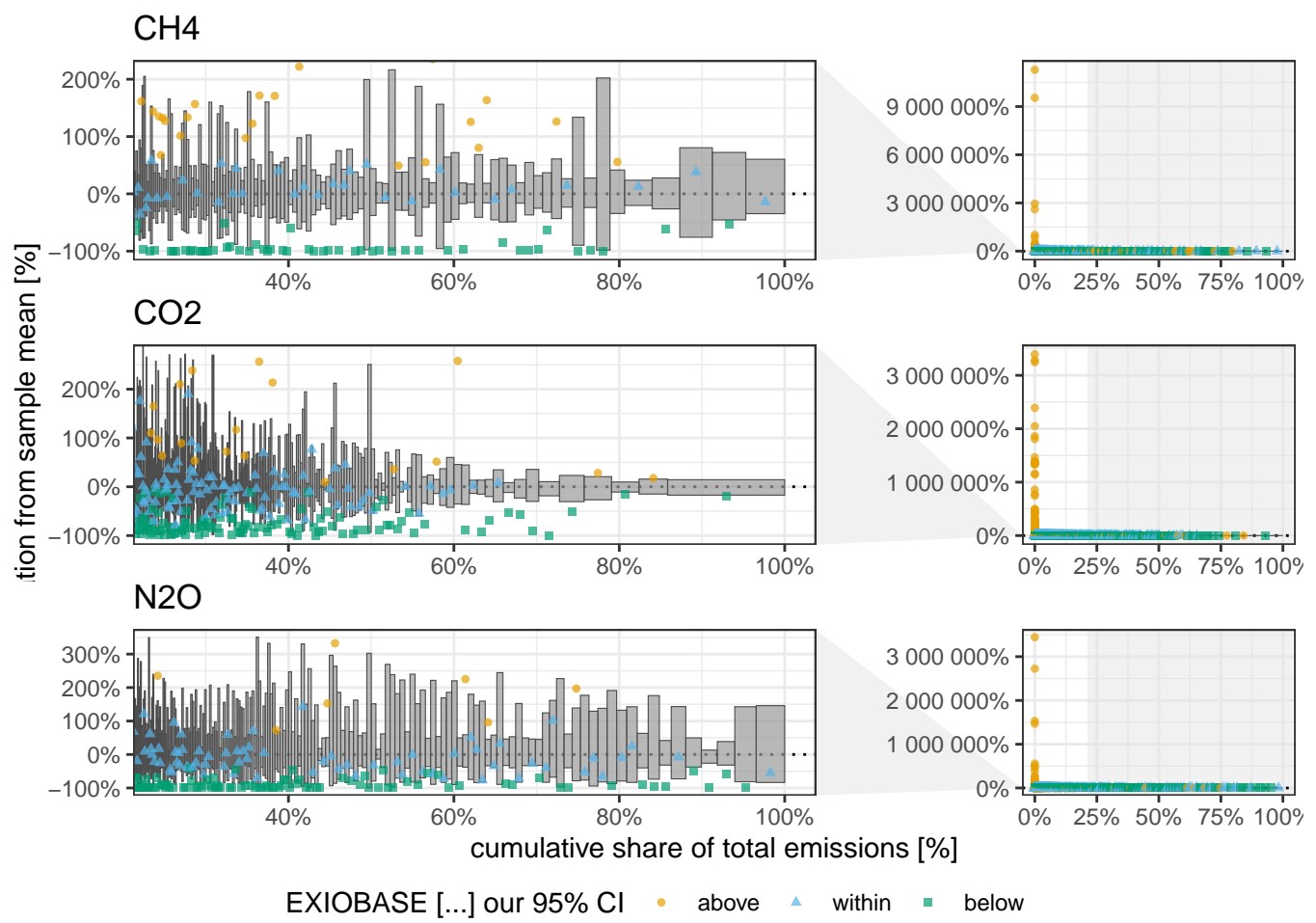

**Figure 7.** Uncertainty of the GHG emissions accounts for the 7987 EXIOBASE sectors against their cumulative share of total emissions. The right column shows all sectors, the left column zooms into those sectors covering the top 80% of total emissions. Uncertainty (y-axis) is shown as the relative deviation from our sample mean. Grey bars display our 95% CI with the width being adjusted to the mean share of total emissions. The coloured points depict the EXIOBASE v3.8.2 estimate for that sector. Colours indicate if the EXIOBASE estimate is above, below or within our 95% CI. Sectors are sorted along the x-axis according to their mean share of total emissions. Note: Sectors with zero emissions are not displayed.





(CH$_4$) and 53% (N$_2$O) of total emissions, the EXIOBASE estimate is below our 95% CI. While sectors for which EXIOBASE
provides higher emission values, make up 14% (CO$_2$), 17% (CH$_4$), and 6% (N$_2$O) of global emissions. Therefore, we conclude
that the uncertainty due to choices outweighs the parametric uncertainty for most sectors, or put differently, all choices made
differently between us and the compilers of the official EXIOBASE GHG emission accounts (Stadler et al., 2018) affect the
sector-level emission accounts more than the raw data uncertainties. Tables E1 and E2 in appendix E1 list all EXIOBASE
sectors for which our sample means are considerably (i.e. for more than 75% of all regions) above or below the official
EXIOBASE V3.8.2 estimate, thus shedding light on sectors for which the process of emission assignment (sec. 2.1) needs
further investigation in terms of proxy data used or incorrect correspondences from raw data items to those sectors.

Comparing the uncertainty in GHG emission accounts with those of the GHG footprint, we see that overall the uncertainty is
substantially lower for the latter. This is indicated by the boxplots in fig. E3 in the appendix, and the summary statistics in tab.
2). From tab. 2 we see that the median CV is factor 5 to 10 lower for the sector-level GHG footprints than for the sector-level
GHG emission accounts. This finding is in line with the results at that country level and can be also explained by the fact that
in the footprint calculations uncertainties are distributed internationally through global supply chains, where they partly cancel
out.

## 4   Discussion

### 4.1   Uncertainty of GHG accounts at the country level

In our analysis, we estimate the uncertainty of the GHG emission accounts by propagating uncertainty from the raw data
inputs needed to compile GHG emissions accounts, namely, the GHG inventories, to the accounts. In doing so, we identify
several uncertainty 'hot-spots'. At the country level, countries with a small economy and comparably little GHG emissions
exhibit higher uncertainties, than large economies with large emissions. A pattern that is strongest for CO$_2$ emissions, in which
case the pattern can mostly be attributed to the residence adjustment (see fig. E2 in appendix E). In the residence adjustment,
large junks of emissions are allocated across many countries. Since we model this allocation using a maximally uninformative
Dirichlet distribution, the amount of emissions attributed to a specific country shows a high variability. For countries like Malta
or Cyprus, where the international aviation and/or shipping sector play a considerably high role in overall economic activities,
this uncertainty also contributes considerably to the uncertainty of the national GHG emission accounts. For large economies
like China or the US, on the other hand, these sectors play - in relative terms - an almost marginal role, therefore, their overall
uncertainty remains relatively unaffected by the uncertainty of the residence adjustment.

Comparing our range estimates with EXIOBASE and OECD point estimates at the country level, we observe that for the
majority of countries, both the EXIOBASE and OECD point estimates are encompassed within our 95% confidence interval.
To conclude, GHG emission accounts at the country level appear reasonably accurate, especially for CO$_2$ emissions, provided
that international transport emissions are a minor component in comparison to the broader economic activities of a country.
However, caution is warranted: disparities between our estimates and those from EXIOBASE and OECD exist for specific
countries (like CH, RO, SE) and some RoW regions, emphasising the importance of choices made while compiling GHG





extensions, such as the source data selection (UNFCCC vs. EDGAR), the approach applied (inventory-first vs. energy-first), the elaboration of correspondences between emission categories and economic sectors, and proxy data selection.

Furthermore, the $CH_4$ and $N_2O$ emission accounts demand a more cautious interpretation. They possess considerably higher
parametric uncertainties, primarily due to the higher uncertainty in raw inventory data (Solazzo et al., 2021). Additionally, the uncertainty resulting from choices, particularly concerning the raw data source (EDGAR vs. UNFCCC), is substantial. For $CH_4$, choosing between EDGAR or UNFCCC can lead to deviations as large as 300% for some countries. For $N_2O$, a clear trend emerges: our method frequently produces higher estimates compared to EXIOBASE/OECD. This recurring variance necessitates further investigation to determine its root causes.

## 4.2   Uncertainty of GHG accounts at the sector level

At the sector level, the uncertainty significantly surpasses that of the country level, with coefficients of variation (CV) reaching values of up to 10. For $CO_2$, uncertainties are distributed unevenly. Larger sectors, in terms of emissions, typically display less uncertainty. However, for $CH_4$ and $N_2O$, the uncertainties are comparatively uniform across industries. Consequently, for $CH_4$ and $N_2O$, even sectors with high emissions exhibit considerable uncertainty. When juxtaposed with the estimates from
EXIOBASE at the sector level, the alignment is less consistent than at the country level. On average, EXIOBASE estimates tend to be below our 95% CIs, while for certain sectors with relatively low emissions, the EXIOBASE estimates show considerable upward deviations.

In conclusion, for GHG emission accounts at the sector level, $CO_2$ estimates for sectors with large emissions seem reasonably accurate. Yet, substantial uncertainties in $CO_2$ emissions exist for sectors with rather low overall emissions, and for $CH_4$ and
$N_2O$ emissions in general. This suggests a more cautious interpretation is warranted for these sectors and emissions. This heightened level of overall uncertainty at the sector level resonates with findings from other studies (Lenzen et al., 2010; Karstensen et al., 2015; Rodrigues et al., 2016), which have also highlighted the need to approach individual data items in a GMRIO database with caution.

Furthermore, the "uncertainty due to choices" (see Huijbregts (1998) and appendix B) is predominant at the sector level.
A significant proportion of the sectors in terms of both, number and emission size, fall outside our 95% CI implying that all choices made differently by us as compared to the EXIOBASE compilers have a greater impact on the sectoral variability than the parametric uncertainty. Consequently, to enhance the robustness of sector-level emission accounts, there is a need for a more systematic analysis on the uncertainty that results from different choices in compiling GHG emission accounts should be made. The aspects needed to consider in such an assessment include industry correspondences, residence adjustments, and the
nature of proxy data utilised (also see appendix E1).

## 4.3   Uncertainty of GHG footprints

In our analysis, we also analyse how the uncertainty propagates further, from the GHG emission accounts, to the GHG footprints. We find that, overall, production-based emission accounts exhibit higher uncertainty than consumption-based accounts (GHG footprints). This is especially pronounced for sectors with high uncertainty in production-based emissions. This finding



can be explained by the fact that when calculating footprints, the production-based emissions along with their uncertainties are distributed internationally through global supply chains to serve final consumption in another part of the world. Since we assume the uncertainties of the production-based emission accounts to be uncorrelated (except those stemming from a common raw data point, see sec. 2.3.2), we expect them to partially cancel out each other, when propagated through the supply chains, resulting in a lower uncertainty of GHG footprints. However, in our analysis we do not include the uncertainty of the entries in

the inter-industry coefficient matrix $\mathbf{A}$, the sectoral output $\mathbf{x}$ and the final demand matrix $\mathbf{Y}$ (see eq. 1). Therefore, depending on the magnitude of these uncertainties and the structure of the correlations we did not cover, consumption-based emission uncertainties might indeed be even higher than the uncertainties from production-based emissions.

### 4.4   Assumptions, limitations and outlook

This analysis provides a conservative baseline scenario of the parametric uncertainty inherent in GHG emission accounts. Thus,

we only - with one exception (see below) - include estimates of raw input data uncertainty, when it comes from authoritative sources and is available to us. In our case, both criteria are fulfilled by two data sources: the National Inventory Reports as submitted to the UNFCCC (Tukker et al., 2018, see), and the study by Solazzo et al. (2021) providing uncertainties for the EDGAR database based on the IPCC 2006 guidelines (IPCC, 2006). For most other raw input data where uncertainty estimates are not available, we assume maximally uninformative distributions. Or, in more technical terms, those distributions

that maximise the statistical entropy. This is especially true for all proxy data used to disaggregate and assign emission inventory data to the MRIO sectors. The only exception, in which we deviate from the Maximum Entropy principle, is for the bridging items used for the residence adjustment concerning international road transport. For these bridging items, we lack a (global) total emission estimate, as compared to international air or water transport emissions. Thus, we need to make an explicit assumption on their uncertainty (see sec. 2.3.1).

Further research could narrow down the uncertainty ranges of raw data inputs for which we used maximally uninformative prior distributions by either relying on expert judgement or using an approach similar to the pedigree matrix approach applied to the life cycle inventory database ecoinvent (Ciroth et al., 2016). This could be achieved by including uncertainty estimates to both, proxy data to assign emissions from international transport (air, ship, fishing) to national GHG accounts in the so-called residence adjustment (sec. 2.1.3), and the proxy data to assign emissions from GHG inventories to MRIO sectors (sec.

2.1.4). The inclusion of uncertainty estimates of the proxy data would be possible in the framework proposed in this study, but replacing the standard Dirichlet distribution as applied here with a generalised Dirichlet distribution such as the one formulated by Plessis et al. (2010) in order to 'force' the disaggregate samples to stay within a given range.

Our analysis comes with several limitations, outlined as follows: In our analysis, following the classification by Huijbregts (1998), we have accounted for two types of uncertainty in GMRIO modelling: parametric uncertainty and uncertainty due to

choices. While in case of the former we have considerably advanced the state-of-the-art in uncertainty estimation in GMRIO, in case of the latter our analysis only provides a very general analysis of the uncertainty due to choices by comparing our GHG emission accounts to other databases which made a different set of choices/assumptions in their compilation process. Particularly in view of our finding that the uncertainty due to choices plays a major role for most sectoral emission and some




country emissions, a more systematic analysis on the uncertainty that results from different choices in compiling GHG emission accounts should be made. This could be achieved with a sensitivity analysis by varying one assumption/choice at a time (e.g. source data, proxy data, sector mapping) to identify the decisions with the largest sources of variability.

Moreover, we neglect other sources of uncertainty and variability, such as model/scenario uncertainty, or spatial variability, temporal variability and variability between objects/sources (see tab. B1 and discussion in appendix B). Especially the latter three sources of variability result in a variability in inputs and outputs (e.g. in forms of GHG emissions) within each GM-RIO sector, which is hidden in the model due to the sector-homogeneity assumption (Majeau-Bettez et al., 2016). While this within-sector variability might be of less relevance at the country-level due to effects of cancelling-out, it might constitute a considerable source of uncertainty for analysis carried out at the sectoral level ("product footprints") or at the sub-national level e.g. in household footprint studies. When interpreting our results, it should be kept in mind that the uncertainty estimates we provide are on the *total emissions* of that sector. However, depending on the characteristics of a sector, the within-sector variability might be substantially larger.

Moreover, in our analysis we also ignore systematic uncertainties arising from limitations of the raw data sources we have used, namely UNFCCC and EDGAR inventories. These limitations include issues with $CO_2$ emissions from biomass combustion as noted by (Pulles et al., 2022), and the predominant use of default emission factors in the case of EDGAR (Crippa et al., 2018). Additionally, there are systematic issues with the proxy data utilised. If proxy data is sourced from the supply table, it neglects all production ending as self-consumption within the same facility and thus not appearing in the supply tables, as for example is the case for coke production used within steel plants. Conversely, using proxy data from the use table fails to account for the re-export by an industry. This means that a sector could use coal without burning it and simply resell it to another sector. In such a scenario, the emissions from that resold coal shouldn't be attributed to that sector but the one that actually burned it.

Furthermore, even uncertainty estimates have their own inherent uncertainty. Although all raw data uncertainty estimates we use in our analysis are based on the same set of guidelines, namely the IPCC 2006 guidelines (IPCC, 2006), there remain questions on comparability and validity. Overall, we consider the uncertainty estimates from the UNFCCC NIRs to be more robust than the ones from EDGAR (Solazzo et al., 2021), since the latter are based on default (tier 1) uncertainty estimates, while the former are often but not exclusively based on more elaborate methods which include national or process-based peculiarities. However, to the best of our knowledge there exist no third-party systematical validation of the UNFCCC NIRs and comparisons with e.g. (Solazzo et al., 2021) to provide more insights into the robustness of the uncertainty estimates.

Finally, our analysis solely captures correlations arising from data disaggregation and omits other potential correlations between the raw data points we use to compile GHG accounts. While for UNFCCC inventories there is no data on correlations available, the uncertainty estimates of the EDGAR inventory from Solazzo et al. (2021) take into account correlations between all emission sources that share the same emission factor. However, in our MC-sampling approach we sample all raw data items independently, thus omitting those correlations. Further research could make use of the correlations from Solazzo et al. (2021) and similarly derive correlations for the UNFCCC inventories by making use of available information on the data structures, e.g. data points that share the same emission factors. However, this "back-engineering" of UNFCCC emission inventories





would require a major effort since the data sources for each emission source (activity data and emission factors) need to be
traced. Therefore, we would encourage the UNFCCC parties to also report correlations along with their uncertainty data. To
make reporting of correlations mandatory and consistent, this could also be reflected in upcoming IPCC guidelines.

## 5   Conclusions

Having robust information on model uncertainty is paramount for robust policy- and decision-making. In this analysis, we
estimated the uncertainty of one part of environmentally-extended GMRIO analysis: the GHG emission accounts. Thereby, we
overcame two major limitations of previous studies: First, instead of making simplistic assumptions, we used authoritative raw
data uncertainty estimates from the National Inventory Reports (NIR) submitted to the United Nations Framework Convention
on Climate Change (UNFCCC) and a recent study on uncertainty of the EDGAR emission inventory. Second, we accounted
for correlations arising from data disaggregation by sampling from Dirichlet distributions.

Our results show a median coefficient of variation (CV) for GHG emission accounts at the country level of 4% for $CO_2$,
12% for $CH_4$, and 33% for $N_2O$. For $CO_2$, smaller economies with significant international aviation or shipping sectors show
CVs as high as 96%, as seen in Malta. At the sector level, uncertainties are higher, with median CVs of 94% for $CO_2$, 100% for
$CH_4$, and 113% for $N_2O$. Overall, uncertainty decreases when propagated from GHG emission accounts to GHG footprints,
likely due to cancelling out effects caused by the distribution of emissions and their uncertainties across global supply chains.
Our GHG emission accounts generally align with official EXIOBASE emission accounts and OECD data at the country level,
though sectoral discrepancies warrant further examination.

To increase the robustness of GHG emission accounts, we recommend future GMRIO compilers the following:

– refine the residence adjustment process to increase robustness of GHG accounts at national level for some small economies

– refine the process of emission assignment in terms of proxy data used and correspondences from emission sources to
   MRIO sectors to address discrepancies in sectoral emissions between our GHG accounts and those from EXIOBASE
v3.8.2 (see tab. E1 and E2 for lists of sectors which are most affected).

To conduct our analysis, a major effort had to be made (in terms of time and resources) to extract and process the uncertainty
estimates from the UNFCCC NIRs which are only published in pdf-format. We made the uncertainty data for the year 2015
(from the 2017 submission) available on Zenodo (Schulte and Heipel, 2023). Moreover, we plan to make data available for
others years as well, however, still some efforts on processing the data is required. In any case, since updating database each year
when the new reports are published requires a consequent effort, we recommend adapting the UNFCCC emissions reporting
guidelines (IPCC, 2006, 2019) to oblige parties to also report their uncertainty estimates in a machine-readable format to make
it easier for researchers to make use of this valuable data resource.

Moreover, we made the correspondence table that maps UNFCCC categories and classifications to EXIOBASE sectors
available on Zenodo (Schulte, 2023). Thereby, we hope to make the refinement of those correspondences, which are a central
piece in compiling GHG emission accounts, a collaborative and cumulative effort.





Lastly, we provide our GHG extensions with associated uncertainties and correlations on Zenodo (Schulte et al., 2023) to complement the official EXIOBASE extensions for users interested in estimating the uncertainties of their results.





## Appendix A: Comparing the GHG emission account compilation between different databases

**Figure A1.** Data sources used to compile selected GHG emission accounts. Own elaboration. Builds on Andrew (2020) (emission data sets), Eurostat (2015) (Eurostat air emission accounts), Flachenecker et al. (2018) (OECD air emission accounts), Stadler et al. (2018) (EXIOBASE air emission accounts), Genty et al. (2012) (WIOD), and Lenzen et al. (2013, 2012) (Eora).



**Appendix B: Sources of uncertainty in GMRIO**

Since, to the best of our knowledge, there exist no proper framework to distinguish different types of uncertainty in GMRIO analysis, we use the classification by Huijbregts (1998) who focus on uncertainty and variability in the related discipline of Life Cycle Assessment (LCA). Distinguishing *variability* from *uncertainty*, Huijbregts (1998) define the former as "stemming from inherent variations in the real world", while the latter is defined as coming "from inaccurate measurements, lack of data, model assumptions, etc.". Variability, as defined by Huijbregts (1998) is sometimes also referred to *aleatoric uncertainty*, while

their understanding of *uncertainty* is often referred to as *epistemic uncertainty* (see e.g. Sullivan (2015)). Huijbregts (1998) further divide (epistemic) uncertainty into parameter uncertainty, model/scenario uncertainty and uncertainty due choices, and variability (aleatoric uncertainty) into spatial variability, temporal variability and variability between objects/sources (Table B1).

**Table B1.** Different types of uncertainty and variability according Huijbregts (1998). GMRIO examples based on own elaboration.

| Type of uncertainty/variability | Description | Example from GMRIO context |
|---|---|---|
| parameter uncertainty | Uncertainty in the outcome caused by uncertainty of input parameters | Uncertainty in economic transaction from sector A to B |
| model/scenario uncertainty | Uncertainty due to (fixed) characteristics of the model structure | Assumption that impacts scale linearly |
| uncertainty due to choices | Uncertainty due to choices that inevitably have to be made in compiling GMRIO databases and using them to calculate environmental impacts | Sectoral or regional resolution |
| spatial variability | Variability across locations | Variability of transport inputs between companies within a sector in a country due to geographic heterogenity |
| temporal variability | Variability in time | Variability of heating inputs between the production of a sector's good at different points in the year |
| variability between objects/sources | Inherent differences in inputs and emissions within a sector in a country | Variability of input structure between companies within a sector in a country due to the use of different technologies |



## Appendix C: EXIOBASE regions

| Country code | Name | Database | Country code | Name | Database |
|---|---|---|---|---|---|
| AT | Austria | UNFCCC | SI | Slovenia | UNFCCC |
| BE | Belgium | UNFCCC | SK | Slovakia | UNFCCC |
| BG | Bulgaria | UNFCCC | GB | United Kingdom | UNFCCC |
| CY | Cyprus | EDGAR | US | United States | EDGAR |
| CZ | Czech Republic | UNFCCC | JP | Japan | EDGAR |
| DE | Germany | UNFCCC | CN | China | EDGAR |
| DK | Denmark | EDGAR | CA | Canada | UNFCCC |
| EE | Estonia | UNFCCC | KR | South Korea | EDGAR |
| ES | Spain | UNFCCC | BR | Brazil | EDGAR |
| FI | Finland | UNFCCC | IN | India | EDGAR |
| FR | France | EDGAR | MX | Mexico | EDGAR |
| GR | Greece | UNFCCC | RU | Russia | EDGAR |
| HR | Croatia | EDGAR | AU | Australia | UNFCCC |
| HU | Hungary | UNFCCC | CH | Switzerland | UNFCCC |
| IE | Ireland | UNFCCC | TR | Turkey | UNFCCC |
| IT | Italy | UNFCCC | TW | Taiwan | EDGAR |
| LT | Lithuania | UNFCCC | NO | Norway | EDGAR |
| LU | Luxembourg | EDGAR | ID | Indonesia | EDGAR |
| LV | Latvia | UNFCCC | ZA | South Africa | EDGAR |
| MT | Malta | UNFCCC | WA | RoW Asia and Pacific | EDGAR |
| NL | Netherlands | UNFCCC | WL | RoW America | EDGAR |
| PL | Poland | UNFCCC | WE | RoW Europe | EDGAR |
| PT | Portugal | UNFCCC | WF | RoW Africa | EDGAR |
| RO | Romania | UNFCCC | WM | RoW Middle East | EDGAR |
| SE | Sweden | UNFCCC | | | |

**Table C1.** EXIOBASE v3 countries/regions and raw data source (EDGAR or UNFCCC inventories). The columns 'Country code' depict ISO 3166-1 alpha-2 codes, except the five Rest of the World (RoW) regions.





## Appendix D: Methods

### D1 International Navigation and Fishing

To estimate the bridging items for international water transport and fishing activities, we use data from Selin et al. (2021) who analysed the allocation of $CO_2$ emission from international shipping to national carbon budgets for the year 2015 using spatially-resolved data on ship movements and ship-specific data on engine power demand, activity time and emissions factor. The authors compared different allocation options: allocation based on flag country, owner country, operator country, manager country or bunker fuel country.

To be consistent with the SEEA we use the allocation based on the operator country since the operator country is where the economic transactions related to the shipping activities are listed in the national accounts (as compared to the flag country that is only responsible for ensuring that a ship meets all relevant legal standards).

To ensure consistency with global international shipping emissions from EDGAR, we do not directly take the bottom-up total emission estimates from Selin et al. (2021). Instead, we take the results from Selin et al. (2021) and calculate country specific use shares by dividing individual countries' shipping $CO_2$-emissions through the sum of all shipping $CO_2$-emissions. Those use shares (which sum to one) are multiplied with the total global international shipping emissions from EDGAR. Formally expressed the emissions of GHG $g$ from international shipping for country $r$ is:

$$E_{r,g} = \frac{E_{r,CO_2}^{Selin2021}}{\sum_{i=1}^{R} E_{i,CO_2}^{Selin2021}} * E_{global,g}^{EDGAR}, \tag{D1}$$

where $E_{r,CO_2}^{Selin2021}$ are all $CO_2$ emissions caused by ships of which the operator is an institutional unit of country $r$ and which serve international shipping or fishing purposes. And $E_{global,g}^{EDGAR}$ are the global $CO_2$/$CH_4$/$N_2O$ emissions related to international shipping from EDGAR.

Since Selin et al. (2021) only cover $CO_2$-emissions, we use the $CO_2$-related use shares for $CH_4$ and $N_2O$, too, thus making the implicit assumption that the ship-specific emission factors for $CH_4$ and $N_2O$ are directly proportional to the $CO_2$-emission factors.

### D2 International Air and Road transport

Unfortunately, for international air and road transport, to the best of our knowledge, there exist no study of a similar scope and level of details as Selin et al. (2021)'s work on international shipping. That's why for both international air and road transport we use the country-specific bridging items provided by Eurostat (Eurostat, 2022).

In case of international road transport we follow Stadler et al. (2018) and consider the EU countries to be by far the most affected by the residence adjustment of international road transport due to the European geography and its economic size (many countries in relatively small area with a lot of cross-border commercial and recreational road transport). Non-European regions represented in EXIOBASE are "either islands or countries with limited road access in relation to their size (e.g. China, India)" (Stadler et al., 2018). Thus we assume that for those countries the road transport emissions from non-resident units operating



on the country's national territory, and the emissions from resident units operating abroad are the same. In other words, we assume the bridging items related to international road transport for non-EU countries to be zero. However, in contrast to international air and water transport we have no knowledge on the total (global/European) emissions from international road transport, as they are not reported separately as memo items but as part of the road transport sector emissions (CRF category

1.A.3.b) within each country. That is why - instead of calculating use shares as we do for water transport - we directly take the total bridging items from Eurostat (Eurostat, 2022) and add/subtract those from the respective EU-country's national road transport emissions.

In case of international air transport, however, non-EU countries constitute a major, definitely non-negligible share of global international air transport emissions. Thus, we calculate country specific use shares separately for EU-countries (using Euro-

stat's bridging data) and non-EU countries. For the latter we use Worldbank data on the country-specific numbers of domestic and international air passengers carried by air carriers registered in the country as a proxy (Worldbank, 2023).

EU-country use shares we calculate as follows:

$$\alpha_{r,g}^{AIR,EU} = \frac{E_{r,g}^{AIR,Eurostat}}{E_g^{AIR,EDGAR}} \tag{D2}$$

Non-EU-country use shares we calculate as follows:

$$\alpha_{r,g}^{AIR,nonEU} = \frac{p_r^{Worldbank}}{\sum_{i \in R} p_i^{Worldbank}} * (1 - \frac{E_{EU}^{Eurostat}}{E_{global}^{EDGAR}}) \tag{D3}$$

**D3   Special case: road transport**

In the first step, we use Eurostat's PEFA data covering fuel-specific energy usage by NACE rev2 industries and households for all EU28 countries plus Iceland and Norway (Eurostat, 2023). We first calculate emissions from the use of "Motor spirit" and "Transport diesel" by industry sector/household $s$ using the default fuel and GHG-specific emission factors $EF_g^{Diesel/Gasoline}$

from IPCC (2006):

$$E_{r,s,g}^{PEFA} = AD_{r,s}^{Diesel} * EF_g^{Diesel} + AD_{r,s}^{Gasoline} * EF_g^{Gasoline}, \tag{D4}$$

where $E_{r,s,g}^{PEFA}$ are the emissions from road transport using PEFA data, $AD_{r,s}^{Diesel/Gasoline}$ is the energy use from Diesel/-Gasoline, both by country $r$ and sector $s$.

By summing emission from both fuel types we get total emissions by industry sector/household. To be consistent with

UNFCCC/EDGAR inventories, we calculate industry/household specific use shares for each country:

$$\alpha_{r,s^{NACE},g}^{ROAD} = \frac{E_{r,s}^{PEFA}}{\sum_{s \in S} E_{r,s}} [ac] \tag{D5}$$



and multiply those with the country specific total emissions from road transport (1.A.3.b) from UNFCCC/EDGAR:

$$E_{r,s^{NACE},g}^{ROAD} = \alpha_{r,s^{NACE},g}^{ROAD} * E_{r,g}^{1.A.3.b} \tag{D6}$$

Next, we allocate emissions from NACErev2 sectors to EXIOBASE sectors analogously using employment data as proxy. For countries not covered by Eurostat's PEFA (all Non-EU countries except Norway and Iceland), we first split the country specific total road transport emissions into household and industry emissions by using the weighted average household-industry split of all countries covered by PEFA, with the total emissions from road transport as weights. We then further disaggregate the industry part using the employment data (see above).

In the case of countries not covered by Eurostat's PEFA (all Non-EU countries excluding Norway and Iceland), our approach involves initially splitting the total road transport emissions into household and industry emissions. To achieve this, we employ a weighted average household-industry split derived from all the countries covered by PEFA, utilising the country specific total emissions from road transport as the weighting factor. Subsequently, we proceed to disaggregate the industry portion by utilising employment data, as mentioned earlier.





# Appendix E: Additional results

## E1    Comparing our estimates to other databases



**Figure E1.** Comparison of the 95% CIs of our GHG emission accounts with the estimates from EXIOBASE and OECD. **(A)**: At country level. **(B)**: at the sector level. Comparison is made according to both, the share of countries/sectors and the share of emissions those countries/sectors generate. Note, due to OECD's broad sector resolution a direct comparison with our emission accounts at the sector level is not possible. In the case of comparison with the OECD estimate, 100% refers to all countries/regions, which are covered both in our analysis and by the OECD.



**Table E1.** Industry sectors for which our sample means are considerably (i.e. for more than 75% of all regions) ABOVE the official EXIOBASE V3.8.2 estimate. Numeric values depict the median, 25%-, and 75%-Quantiles, respectively, of the sector-wise relative differences between our sample mean and the official EXIOBASE V3.8.2 estimate. A median of 19, for example, means that the median relative difference for that specific industry sector among all 49 regions is factor 19 compared to the official EXIOBASE estimates. CO2 only.

| Industry name | Industry code | Median | $Q_{0.25}$ | $Q_{0.75}$ |
|---|---|---|---|---|
| Re-processing of secondary preciuos metals into new preciuos metals | i27.41.w | 33.03 | 2.88 | 20841.48 |
| Production of electricity by nuclear | i40.11.c | 31.76 | 0.83 | 433.56 |
| Other land transport | i60.2 | 18.97 | 2.23 | 914.90 |
| Cultivation of crops nec | i01.h | 7.15 | 0.97 | 22.42 |
| Manufacture of rubber and plastic products (25) | i25 | 4.48 | 0.85 | 13.54 |
| Processing vegetable oils and fats | i15.e | 4.08 | 0.22 | 15.46 |
| Manufacture of machinery and equipment n.e.c. (29) | i29 | 3.26 | 1.28 | 7.13 |
| Biogasification of food waste, incl. land application | i90.2.a | 3.25 | 0.33 | 8.68 |
| Wholesale trade and commission trade, except of motor vehicles and motorcycles (... | i51 | 2.75 | 1.04 | 6.91 |
| Construction (45) | i45 | 2.57 | 0.78 | 6.14 |
| Biogasification of sewage slugde, incl. land application | i90.2.c | 2.57 | 1.13 | 7.04 |
| Retail trade, except of motor vehicles and motorcycles; repair of personal and h... | i52 | 2.53 | 0.48 | 6.48 |
| Sale, maintenance, repair of motor vehicles, motor vehicles parts, motorcycles, ... | i50.a | 2.07 | 0.44 | 7.21 |
| Other service activities (93) | i93 | 1.94 | 0.73 | 6.33 |
| Other business activities (74) | i74 | 1.83 | 0.66 | 4.08 |
| Poultry farming | i01.k | 1.12 | 0.33 | 3.82 |
| Forestry, logging and related service activities (02) | i02 | 0.94 | 0.24 | 6.01 |
| Production of electricity by petroleum and other oil derivatives | i40.11.f | 0.81 | 0.06 | 6.75 |

## E2   Decomposing the uncertainty by source sector

Compiling GHG emission accounts from UNFCCC and EDGAR emission inventories involves allocating each emission source reported in the inventories to EXIOBASE target sectors and countries (see sec. 2.1). Both, UNFCCC and EDGAR report their emissions in the CRF format.

Here, we aggregate all allocations by CRF category at a common level (depicting the emission source) and at the EXIOBASE region level (depicting the emission destination) to analyse which source categories contribute most on overall uncertainty by country. Figure E2 illustrates the absolute uncertainty by CRF category and country using bubble charts. The size of each bubble is adjusted according to the share of the standard deviation (SD) of the respective IPCC category in the sum of all SDs for each country. While interpreting the actual numbers, caution is advised since summing all SDs may not yield a meaningful

calculation. Nevertheless, the visualisation remains helpful in identifying 'hot-spots' of uncertainty sources.





**Figure E2.** Breakdown of the uncertainty by CRF category and country using bubble charts. The size of each bubble is adjusted according to the share of the standard deviation (SD) of the respective IPCC category in the sum of all SDs for each country.





**Table E2.** Industry sectors for which our sample means are considerably (i.e. for more than 75% of all regions) BELOW the official EX-IOBASE V3.8.2 estimate. Numeric values depict the median, 25%-, and 75%-Quantiles, respectively, of the sector-wise relative differences between our sample mean and the official EXIOBASE V3.8.2 estimate. A median of -0.9, for example, means that the median relative difference for that specific industry sector among all 49 regions is factor -0.9 compared to the official EXIOBASE estimates. CO2 only.

| Industry name | Industry code | Median | $Q_{0.25}$ | $Q_{0.75}$ |
|---|---|---|---|---|
| Production of electricity by tide, wave, ocean | i40.11.j | NA | -1.00 | -0.79 |
| Production of electricity by solar thermal | i40.11.i | -1.00 | -1.00 | -0.99 |
| Re-processing of ash into clinker | i26.d.w | -1.00 | -1.00 | -1.00 |
| Casting of metals | i27.5 | -0.98 | -0.99 | -0.93 |
| Production of electricity by solar photovoltaic | i40.11.h | -0.98 | -1.00 | -0.65 |
| Retail sale of automotive fuel | i50.b | -0.96 | -0.99 | -0.79 |
| Re-processing of secondary glass into new glass | i26.a.w | -0.94 | -0.98 | -0.89 |
| Manufacture of gas; distribution of gaseous fuels through mains | i40.2 | -0.89 | -0.98 | -0.38 |
| Mining of aluminium ores and concentrates | i13.20.13 | -0.89 | -0.98 | -0.17 |
| Inland water transport | i61.2 | -0.86 | -0.97 | -0.66 |
| Mining of iron ores | i13.1 | -0.86 | -1.00 | -0.27 |
| Extraction of natural gas and services related to natural gas extraction, exclud... | i11.b | -0.85 | -0.96 | -0.45 |
| Manufacture of other non-metallic mineral products n.e.c. | i26.e | -0.54 | -0.77 | -0.09 |

Regarding $CO_2$ emissions, most countries exhibit the largest SD in the allocations of international air and water emissions. Additionally, road transportation (1.A.3) and other energy sources (1.A.4) also contribute significantly to the overall uncertainty in some countries. For $CH_4$ emissions, the greatest uncertainty arises from Agriculture (3) and Waste (5) with varying compositions depending on each country. Fugitive emissions (1.B) are generally less pronounced but still constitute a major source in some countries. Notably, for a few countries like Malta, Cyprus, Denmark, and Luxembourg, $CH_4$ emissions from international water transport emerge as a significant source of uncertainty.

Concerning $N_2O$ emissions, emissions from agriculture (3) account for the largest uncertainty in almost all countries. However, for a few countries like Spain, Taiwan, Cyprus, and Malta, emissions from waste treatment (5) and international shipping (0.B) also play a considerable role.

## E3 Uncertainty at sectoral level

Figure E3 shows the uncertainty in GHG emission accounts (left column) next to the uncertainty of the GHG footprints (right column). To allow comparison between the two, the measure of uncertainty is reduced to one number, the CV (y-axis, orange line). Each coloured stripe represents one sector with the width and colour adjusted to the mean share in total emissions. In the case of GHG emission accounts the mean share refers to the (total) production-based emissions of that sector, while in case of

**Figure E3.** The relative standard error (CV) of the GHG emission accounts at the sectoral level (coefficients) (orange line, y-axis) by the cumulative share of total emissions (x-axis). Each coloured stripe represents one sector, with the width and colour adjusted to the mean share of total emissions. The sectors are sorted along the x-axis by their CV from high (left) to low (right). The width and colour of the stripes is adjusted to the mean share of total emissions. To ease interpretation, the dotted line marks a CV of 0.1 and the dashed line a CV of 0.5. The boxplots show the (unweighted) distributions of the CV of the individual sectors.

the GHG footprints the mean share refers to the consumption-based emissions to satisfy *global* final demand of that sector's product. The stripes are sorted along the x-axis by their CV. The boxplots show the (unweighted) distributions of the CV of the individual sectors.

Focusing on the uncertainty in GHG emissions accounts (fig. E3, left column), we see that the distributions of the CV of the sectors look very similar between the three GHGs (fig. E3 boxplots, tab. 2 for values). However, in case of $CO_2$ large

uncertainties mostly occur with sectors with a relatively low share in total emissions, while sectors with relatively high $CO_2$ emissions show in general a lower uncertainty. This can be seen in the distribution of the width and colours along the y-axis



in fig. E3, where for $CO_2$ sectors with high emissions (large stretch along x-axis, yellow to green colour) are generally more situated towards the right (lower CV), while sectors with low $CO_2$ emissions (small stretch along x-axis, blue colour to purple) are more present towards the left (high CV). For $CH_4$ and $N_2O$ there is no clear trend between a sector's CV and its size in terms of emissions. Thus, high-emitting sectors (wide, yellow boxes) and low-emitting sectors (narrow, blue boxes) are much more evenly distributed along the y-axis. This finding aligns well with what we saw in fig. 7, namely that uncertainties $CH_4$ and $N_2O$ emissions are much more homogeneously distributed between sectors than in case of $CO_2$. We see that for $CO_2$ 37% of all emissions stem from sectors with a CV of less than 0.1 (dashed line) and 85% from sectors with a CV of less than 0.5 (dotted line). While for $CH_4$ it is only 7% (CV < 0.1) and 80% (CV < 0.5), and for $N_2O$ only 2% (CV < 0.1) and 41% (CV < 0.5).





**Code and data availability**

- Our GHG extensions with associated uncertainties and correlations is made available on Zenodo under https://doi.org/10.5281/zenodo.10037713 (Schulte et al., 2023)

- The extracted and processed uncertainty data from the UNFCCC NIRs (submission 2017) is made available on Zenodo under https://doi.org/10.5281/zenodo.10037714 (Schulte and Heipel, 2023).

- Our correspondence table that maps UNFCCC CRF categories and classifications to EXIOBASE sectors is made available here: https://doi.org/10.5281/zenodo.10046372 (Schulte, 2023).

- The R-code needed to reproduce the results of this article is available on Github and Zenodo: https://doi.org/10.5281/zenodo.10141616.
- EXIOBASE v3.8.2 which was used in this analysis is available under https://doi.org/10.5281/zenodo.5589597 (Stadler et al., 2021).

*Video supplement.* use this section when having video supplements available

*Author contributions.* SS, SP and AJ conceptualised and designed the research. SS performed the computations and analysed the results. SS, SP and AJ validated the results. SS and wrote the paper with inputs from all authors.

*Competing interests.* The authors declare no competing interests.

*Acknowledgements.* We thank Joshua Heipel for helping with the extraction of the uncertainty data from the UNFCCC National Inventory reports.



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
