# Peer review of "Estimating the uncertainty of the greenhouse gas emission accounts in Global Multi-Regional Input-Output analysis"

_Earth System Science Data, 2023_

## Author Comment (AC3)

**Reviewer 1**

The authors did a great job, they processed a huge amount of badly structured information from the national inventory reports in pdf format and other datasets. The data they compiled and estimated on uncertainties of the National GHG emission estimates submitted by countries to the UNFCCC and for the Global Multi-Regional Input-Output analysis are new, at least I did not find similar data. The data are useful for researchers as well as for policy makers because knowing the uncertainties is as important as knowing the values themselves.

**General reply**

We thank the reviewer for their positive evaluation, as well as for their constructive and valuable comments. See below for our replies on your specific comments, including a short explanation how we addressed them.

The manuscript contains a very detailed description of the method, results, and a good discussion of limitations. However, the manuscript contains so many details (e.g. explanation of what is Annex-I parties) that sometimes it is challenging to track the most important information. I would suggest moving all the auxiliary information to the footnotes. Otherwise, everything is described very well.

Thank you for your suggestion. We streamlined parts of the data description section by leaving out information that is not directly relevant for the understanding of our methods. We have refrained from using footnotes, however, as the ESSD submission guidelines state that "Footnotes should be avoided in the text".

I have two specific comments:

Could you please clarify what is WM in paragraph 485?

Throughout the results and discussion sections we switched from using country codes to using their names to make it easier for the reader to follow the text. At first occurrence we additionally put the codes in brackets to facilitate the connection of text and plots (see lines 490ff).

Could you please correct the legend in Figure E2 (p.38)?

Thanks for spotting. We adjusted the legend (and the axis labels).

Data quality.

Unfortunately, I failed with running the R script developed by the authors. I could not find the package mRio. Probably, it is my fault as I'm not an experienced user of R.

We have responded to this comment in the first answer (AC1 from 19th of Feb 2024) and hope that the reviewer managed to run the code with the adjustments made.

I looked at some of the data separately. However, it would be good if the authors developed a script for exploring the data that does not require special packages.

We are not sure if we fully understand the reviewer's request. Our results data is provided on Zenodo in the '.feather' format which allows sharing data between different platforms (Python, R, C, etc.). So yes, the reviewer is right that to read 'feather' data you need a special package (e.g. "arrow" in R, or "PyArrow" in python). However, we decided to take the 'feather' format for performance reasons. Writing and reading feather files is orders of magnitude faster than e.g. csv files.

I could not estimate the quality of the data comprehensively. I put **major revisions** because of my problem with the R script and, as a result, missing comprehensive data quality assessment.
* * *
**Reviewer 2**

This is a good paper. Uncertainties in GHG accounts (production- and consumption-based) is hot topic and very high on the agenda of modellers and decision makers alike. The authors do a great job in shedding light on various issues. I was not aware that uncerainty estimates of the UNFCCC data set did exist in the NRI. This paper is clearly contributing to leveraging this infromation and providing it to a wider audience. The findings are relevant in highlighting the sectors and regions that are of concerne. Using Dirichlet distributions to capture correlations in data disaggregation is a smart move. Using entropy approaches in setting the paramters of the Dirichlet distribution is even smarter. I like that approach. The GHG extensions and uncertainties and meta data provided by the authors is very useful. I suggest this paper for publication after a minor revision.

**General reply**

We thank the reviewer for their positive and kind evaluation, as well as for their constructive and valuable comments. See below for our replies on your specific comments, including an explanation how we addressed them.

1. So how did the EXIOBASE team construct the GHG extensions? I am sure the answer to that question is already contained in the manuscript but I would love to see this stated more clearly. Is the EXIOBASE extension based on UNFCCC alone? This is not clear to me after reading the text and when MC results are compared to the original EXIOBASE extension are was wondering what I am lookig at here.

That is a good point. We had a very (too) short description on this in the original manuscript. In the revised version (line 195ff) we included more details on the how EXIOBASE compiled their GHG extensions:

"Thus, our approach differs from the EXIOBASE approach as of version 3.8.2. EXIOBASE compilers apply the energy-first by using IEA World Energy Balances to compile energy use accounts and then combine those energy use data with emission factors from the TEAM model (Stadler et al., 2018; Pulles et al., 2007). Thereafter, $CO_2$ fossil emissions (until 2019) at an aggregate level are scaled to match EDGAR emissions and all other GHG emissions (until 2017) scaled to match the PRIMAP database (see "README" files in Stadler et al. (2021)). In appendix A, we provide a figure on the different data sources used to compile selected GHG emission accounts and how they differ (fig. A1)."

Moreover, at the respective places in the result section we link to that description to make it easier for the reader to make sense of the comparisons of our results with official EXIOBASE numbers.

2. You have all this information on uncertainties. Would it be possible for you to suggest a sector classification for GHG footprins from EXIOBASE that some incorporates that insights on uncertainties. I mean you implicitly say that assessing GHG footprints on the level of 163 sectors, is for many sector not a good idea due to the uncertainties. And yes, I am aware that such a decision depends greatly on the research question. However, if one aims at an analysis of the GHG footprints of sectors for a given country, what would be a robust sector aggregation that somehow "outsources" or "collects" the more problematic sectors in a lets call it "highly uncertain" sector group (something like products nec). This would be of high value for analysts. Maybe that is beyond the scope of your paper, but I think you are highly qualified to give such a suggestion or at least some reflections.

That is a very good, but also tricky question. First of all, as shown by Lenzen (2011) you get more accurate results if you first disaggregate, then do the calculations, and then aggregate the results. Therefore, for database compilers, we would still recommend going for a higher level of detail. For database users and analysts, however, we totally agree with the referee, that there is a need for more guidance on the "right" level of aggregation of the results.

One thing to keep in mind when it comes to the question of the "best" level of aggregation is that there is always a trade-off between the aggregation error and the uncertainty on the mean. The more you aggregate sectors, the lower the uncertainty on the mean (due to cancelling out effects, except when uncertainties are highly positively correlated), but the higher the aggregation errors.

Having a sector classification that is more in line with UNFCCC categories would most certainly decrease sector-level uncertainty on the means due to less disaggregation required. However, this would probably come at the cost of higher uncertainties of the economic transactions since, as mentioned in our manuscript (lines 305ff), emission inventories have a more technical process-oriented classifications, while the economic IO tables have a classification based on economic activity. Thus we would need to introduce more assumptions (and therewith uncertainties) to the align the IO data to the UNFCCC category classification. But, yes this would be an interesting exercise to conduct.

A more thorough analysis could involve looking for sectors with similar average emission multipliers and aggregate them to one sector and check if the uncertainty on the mean decreases. This kind of analysis is definitely feasible using the raw MC samples from our study, but would make another paper. Especially since, as mentioned by the referee, the level of aggregation of the results heavily depends on the research question.

We added a paragraph to the discussion section (line 680):

"Moreover, there is a trade-off between the within-sector variability and the uncertainty of the mean with respect to the sectoral resolution: the more you aggregate sectors, the lower the uncertainty on the mean (due to cancelling out effects, except when uncertainties are highly positively correlated), but the higher the within-sector variability. As shown by Lenzen (2011), IO-based results are more accurate if you first disaggregate the IO data, then perform the calculations, and then aggregate the results. Thus, we still recommend that database compilers to further increase the sectoral resolution of GMRIO databases to decrease the aggregation bias, even though we expect an even higher uncertainty on the mean than what we found here for the EXIOBASE resolution. For database users and analysts, however, our analysis indicates that there is a need for more guidance on the "best" level of aggregation of GMRIO-based results that balances out the aggregation error with the uncertainty on the mean. Yet, we leave this research topic which greatly depends on the research question, for further research"

3. You provide a good overview on the different approaches to estimating uncertainties of the production-based emissions accounts (heuristics vs. modelling with power series regressions): Am I right in assuming that all power series regressions actually rely on the "law of large numbers"? Moreover, looking the MC results for sectors and countries, it seems to me you arrive at similar findings that could be understood as supporting the law of large numbers assumption: The larger a sector or region the smaller the uncertainty in relative terms. I don't want you to reframe the introduction but I was wondering whether to a ceratin degree the law of large numbers and more simplistic approach are actually justified. Would be interesting to see some reflections how your findings connect to that.

For your first question, yes, you are correct, all studies we review that apply power series regressions rely on the "law of large numbers".

We agree that for $CO_2$ (only), that there is a negative correlation between national/sector footprints and relative uncertainties. Thus, our results confirm that for $CO_2$ as a rule of thumb the law of large numbers is a reasonable assumption. However, our results also show that the law of large numbers only explains a (small) part of the overall variability in the uncertainties.

Inspired by your comment, we added a section and a figure in the appendix to examine how the "law of large numbers" performs on our **raw** data uncertainty. (see lines 830ff) There we show that the sector size mostly explains - if at all - a (small) part of the overall variability in uncertainties.

Moreover, we added a paragraph in the discussion (line 705):

"Yet, despite those limitations, by using "real" uncertainty estimates on the raw data instead of modelling them using a power law regression or heuristics (as previously done), we consider our analysis to substantially advance uncertainty modelling in GMRIO. Especially since the "law of large numbers" mostly explains only a small part of the overall variability in uncertainties (see sec. E1n the appendix)."

And to the method section (line 405):

"To examine how the "law of large numbers" which has been used the literature (Lenzen et al., 2010; Karstensen et al., 2015; Zhang et al., 2019) performs for the uncertainty data from the UNFCCC and EDGAR, we fit power law regressions both pooled and by-country (fig. E1 and sec. E1 in the appendix)."

4. I am confused by the terms "classification" and "categories" in the context of the UNFCCC data. See Figure 3 for example. Does category stand for process/industry and classification for the type GHG emissions i.e. flow? I am confused. Please clarify that somehow a bit better.

We introduce and explain both terms in paragraph (lines 221ff). However, we acknowledge that due to the length of the article, definitions and explanations of terms can easily be missed and/or hard to find.

Therefore, we modified the caption of Figure 3 so that it now reads: "An example of the nested hierarchical data structure of the UNFCCC inventories: One hierarchy on the left side representing the categories (e.g. 1.A.2 represents emissions from "Manufacturing Industries and Construction"). Each node representing a category contains another hierarchy representing the classification (i.e. fuel/animal type)."

5. You make a great job in detailing the importance of how to allocate international road transport, especially for european countries. I would love to read a bit more about that in your reflections. Is there something you would recommend to modellers that are dealing with the same problem (best practice)? This would be a nice to have, not a must-have. I think this is really another important intervention point for dealing with uncertainties in the future, which deserves a bit more attention.

We agree, the residence adjustment is an important contributor to overall uncertainty because very large junks of emissions are distributed to countries with (what we assume) little information on the robustness of those allocations.

Thus, yes, a more thorough and standardised approach would be a great contribution to more robust MRIO models. However, we think that based on our analysis, we can only make the point, that the residence adjustment *should* be made (for the sake of consistency with the SNA) and that we should get better proxies/models for doing it (i.e. proxies from which we know that they have are relatively accurate). We think that those points become very clear from our results, discussion and conclusion.

However, we do not feel qualified enough to suggest a best-practice for how to do it (i.e. which proxies/model you use), because this requires a broader analysis and testing, which was not within the scope of our paper.

The DOI link you give to your GHG extensions (https://doi.org/10.5281/zenodo.10037713 ) is forwarding me to the UNFCCC uncertainty data set (https://zenodo.org/records/10037714 ). Is this meant to be like that? Please check the links.

Thanks for spotting. We corrected the link.

All in all, a well written and interesting piece of work. I congratulate the authors.

Thank you very much again for your constructive comments.